# Exploring the Impact of Information Entropy Change in Learning Systems

## Abstract

We explore the impact of entropy change in deep learning systems via noise injection at different levels, i.e., the latent space and input. The applications in this work focus on supervised learning tasks within computer vision, but the proposed method can be further adapted to other fields. Noise is conventionally viewed as a harmful perturbation in various deep learning architectures, such as convolutional neural networks (CNNs) and vision transformers (ViTs), as well as different learning tasks like image classification and transfer learning. However, this work shows noise can be an effective way to influence the entropy of the learning systems. We demonstrate that specific noise can boost the performance of various deep architectures under certain conditions. We theoretically prove the enhancement gained from positive noise by reducing the task complexity defined by information entropy and experimentally show the significant performance gain in large image datasets, such as the ImageNet. Herein, we use the information entropy to define the complexity of the task. We categorize the noise into two types, positive noise (PN) and harmful noise (HN), based on whether the noise can help reduce the complexity of the task. Extensive experiments of CNNs and ViTs have shown performance improvements by proactively injecting positive noise, where we achieved an unprecedented top 1 accuracy of over 95% on ImageNet. Both theoretical analysis and empirical evidence have confirmed that the presence of positive noise can benefit the learning process, while the traditionally perceived harmful noise indeed impairs deep learning models. The different roles of noise offer new explanations for deep models on specific tasks and provide a new paradigm for improving model performance. Moreover, it reminds us that we can influence the performance of learning systems via information entropy change.

## 1 Introduction

In supervised learning tasks, models are carefully crafted to improve the quality of representations and mappings between input data and their corresponding labels. A well-trained model can significantly reduce task uncertainty, thus providing predictions with a remarkable degree of confidence throughout the training process Van Engelen & Hoos (2020). Fortunately, the concept of entropy from information theory offers an excellent framework for illustrating uncertainty and, consequently, offers insights into enhancing the performance of learning systems through entropy manipulation Shannon (1948). A straightforward but effective way to influence the entropy of the learning systems is to add noise Li (2022). Noise, conventionally regarded as a hurdle in machine learning tasks, is universal and unavoidable in science and engineering tasks, most works assume that noise has a negative impact on the current task Sethna et al. (2001) Owotogbe et al. (2019). However, noise can actually help reduce the task complexity that is defined by information entropy and positively influence learning systems Radnosrati et al. (2020) Zhang et al. (2023).

We recognize that task complexity is the essential concept in our work that needs to be explicitly introduced in the first place. We utilize task complexity to characterize a learning system. High task complexity signifies greater uncertainty, indicating suboptimal performance, while low task complexity denotes lower uncertainty, signifying a high-performance learning system. The task complexity is measured by the information entropy, rooted in the information theory. By using the definition of task complexity, it is possible to categorize noise into two distinct categories: **positive**

**noise (PN)** and **harmful noise (HN)**. PN decreases the complexity of the task, while HN increases it, aligning with the conventional understanding of noise.

Our work aims to comprehensively investigate how various types of noise affect task complexity and impact learning system performance. Specifically, the study focuses on three common types of noise, i.e., Gaussian noise, linear transform noise, and salt-and-pepper noise. Gaussian noise refers to random fluctuations that follow a Gaussian distribution in pixel values at the image level or latent representations in latent space Russo (2003). Linear transforms, on the other hand, refer to affine elementary transformations to the latent representations or input images, where the transformation matrix is row equivalent to an identity matrix Marcus & Moyls (1959). Salt-and-pepper noise is a kind of image distortion that adds random black or white values at the image level or to the latent representations Chan et al. (2005).

This paper analyzes the impact of these types of noise on the performance of deep learning models for image classification and domain adaptation tasks. Two popular model families, Vision Transformers (ViTs) and Convolutional Neural Networks (CNNs), are considered in the study. Image classification is one of the most fundamental tasks in computer vision, where the goal is to predict the class label of an input image. Domain adaptation is a practically meaningful task where the training and test data come from different distributions, also known as different domains. By investigating the effects of different types of noise on ViTs and CNNs for typical deep learning tasks, the paper provides insights into the influences of noises on deep models. The findings presented in this paper hold practical significance for enhancing the performance of various types of deep learning models in real-world scenarios.

The contributions of this paper are summarized as follows:

- We re-examined the conventional view that noise, by default, has a negative impact on deep learning models. Our theoretical analysis and experimental results show that noise can be a positive support for deep learning tasks.

- We conducted extensive experiments using various deep models, including CNNs and ViTs, across different deep learning tasks such as classification and domain adaptation. By leveraging positive noise, we attained state-of-the-art (SOTA) results.

- We theoretically analyze the distinction between injecting noise at the image level and in the latent space. Our findings reveal that injecting noise in the latent space significantly reduces task complexity compared to image-level operations.

- The theory and framework of reducing task complexity via positive noise in this work can be applied to any deep learning architecture. There is great potential for exploring the application of positive noise in other deep-learning tasks beyond the image classification and domain adaptation tasks examined in this study.

## 2    RELATED WORK

**Positive Noise** In fact, within the signal-processing society, it has been demonstrated that random noise helps stochastic resonance improve the detection of weak signals Benzi et al. (1981). Noises can have positive support and contribute to less mean square error compared to the best linear unbiased estimator Radnosrati et al. (2020). Also, it has been reported that noise could increase the model generalization in natural language processing (NLP) Pereira et al. (2021). Recently, the perturbation, a special case of positive noise, has been effectively utilized to implement self-refinement in domain adaptation and achieved state-of-the-art performance Sun et al. (2022). The latest research shows that by proactively adding specific noise to partial datasets, various tasks can benefit from the positive noise Li (2022). Besides, noises are found to be able to boost brain power and be useful in many neuroscience studies McClintock (2002) Mori & Kai (2002).

**Deep Model** Convolutional Neural Networks have been widely used for image classification, object detection, and segmentation tasks, and have achieved impressive results LeCun & Bengio (1995)He et al. (2016). However, these networks have limitations in terms of their ability to capture long-range dependencies and extract global features from images. Recently, Vision Transformers has been proposed as an alternative to CNNs Dosovitskiy et al. (2020). ViT relies on self-attention mechanisms and a transformer-based architecture to enable global feature extraction and modeling

of long-range dependencies in images Vaswani et al. (2017). ViT has demonstrated impressive performance on a range of image classification tasks and has the potential to outperform traditional CNN-based approaches.

## 3   PRELIMINARY

In information theory, the entropy Shannon (2001) of a random variable $x$ is defined as:

$$H(x) = \begin{cases} -\int p(x)\log p(x)dx & \text{if } x \text{ is continuous} \\ -\sum_x p(x)\log p(x) & \text{if } x \text{ is discrete} \end{cases} \tag{1}$$

where $p(x)$ is the distribution of the given variable $x$. The mutual information (MI) of two random discrete variables $(x, y)$ is denoted as Cover (1999):

$$\begin{aligned} MI(x,y) =& D_{KL}(p(x,y) \parallel p(x) \otimes p(y)) \\ =& H(x) - H(x|y) \end{aligned} \tag{2}$$

where $D_{KL}$ is the Kullback–Leibler divergence Kullback & Leibler (1951), and $p(x, y)$ is the joint distribution. The conditional entropy is defined as:

$$H(x|y) = -\sum p(x,y)\log p(x|y) \tag{3}$$

The above definitions can be readily expanded to encompass continuous variables through the substitution of the sum operator with the integral symbol. In this work, the noise is denoted by $\epsilon$ if without any specific statement.

Before delving into the correlation between task and noise, it is imperative to address the initial crucial query of the mathematical measurement of a task $\mathcal{T}$. With the assistance of information theory, the complexity associated with a given task $\mathcal{T}$ can be measured in terms of the entropy of $\mathcal{T}$. Therefore, we can borrow the concepts of information entropy to explain the difficulty of the task. For example, a smaller $H(\mathcal{T})$ means an easier task and vice versa.

Since the entropy of task $\mathcal{T}$ is formulated, it is not difficult to define the entropy change when additional noise $\epsilon$ is present Li (2022),

$$\triangle S(\mathcal{T}, \epsilon) = H(\mathcal{T}) - H(\mathcal{T}|\epsilon) \tag{4}$$

Formally, if the noise can help reduce the complexity of the task, i.e., $H(\mathcal{T}|\epsilon) < H(\mathcal{T})$ then the noise has positive support. Therefore, a noise $\epsilon$ is defined as **positive noise** (PN) when the noise satisfies $\triangle S(\mathcal{T}, \epsilon) > 0$. On the contrary, when $\triangle S(\mathcal{T}, \epsilon) \leq 0$, the noise is considered as the conventional noise and named **harmful noise** (HN).

$$\begin{cases} \triangle S(\mathcal{T}, \epsilon) > 0 & \epsilon \text{ is positive noise} \\ \triangle S(\mathcal{T}, \epsilon) \leq 0 & \epsilon \text{ is harmful noise} \end{cases} \tag{5}$$

**Moderate Model Assumption**: The positive noise may not work for deep models with severe problems. For example, the model is severely overfitting where models begin to memorize the random fluctuations in the data instead of learning the underlying patterns. In that case, the presence of positive noise will not have significant positive support in improving the models' performance. Besides, when the models are corrupted under brute force attack, the positive noise also can not work.

## 4   METHODS

The idea of exploring the influence of noise on the deep models is straightforward. The framework is depicted in Fig. 1. This is a universal framework where there are different options for deep models, such as CNNs and ViTs. Through the simple operation of injecting noise into a randomly selected layer, a model has the potential to gain additional information to reduce task complexity, thereby improving its performance. It is sufficient to inject noise into a single layer instead of multiple layers since it imposes a regularization on multiple layers simultaneously.

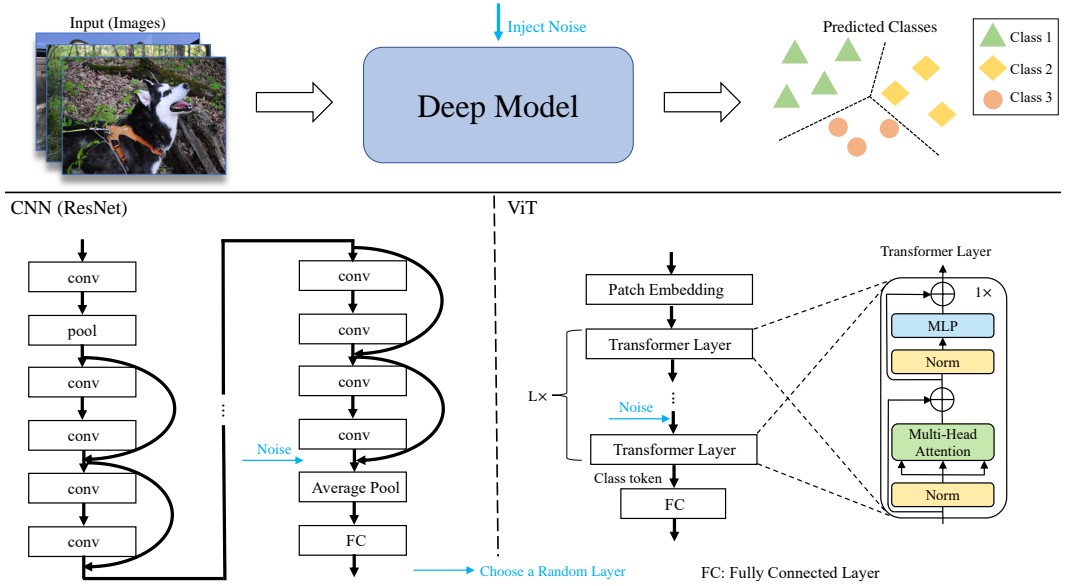

Figure 1: An overview of the proposed method. Above the black line is the standard pipeline for image classification. The deep model can be CNNs or ViTs. The noise is injected into a randomly chosen layer of the model represented by the blue arrow.

For a classification problem, the dataset $(\boldsymbol{X}, \boldsymbol{Y})$ can be regarded as samplings derived from $D_{\mathcal{X}, \mathcal{Y}}$, where $D_{\mathcal{X}, \mathcal{Y}}$ is some unknown joint distribution of data points and labels from feasible space $\mathcal{X}$ and $\mathcal{Y}$, i.e., $(\boldsymbol{X}, \boldsymbol{Y}) \sim D_{\mathcal{X}, \mathcal{Y}}$ Shalev-Shwartz & Ben-David (2014). Hence, given a set of $k$ data points $\boldsymbol{X} = \{X_1, X_2, ..., X_k\}$, the label set $\boldsymbol{Y} = \{Y_1, Y_2, ..., Y_k\}$ is regarded as sampling from $\boldsymbol{Y} \sim D_{\mathcal{Y}|\mathcal{X}}$. The complexity of $\mathcal{T}$ on dataset $\boldsymbol{X}$ is formulated as:

$$H(\mathcal{T}; \boldsymbol{X}) = H(\boldsymbol{Y}; \boldsymbol{X}) - H(\boldsymbol{X}) \tag{6}$$

Accordingly, the operation of adding noise at the image level can be formulated as Li (2022):

$$\begin{cases} H(\mathcal{T}; \boldsymbol{X} + \boldsymbol{\epsilon}) = -\sum_{\boldsymbol{Y} \in \mathcal{Y}} p(\boldsymbol{Y}|\boldsymbol{X} + \boldsymbol{\epsilon}) \log p(\boldsymbol{Y}|\boldsymbol{X} + \boldsymbol{\epsilon}) & \boldsymbol{\epsilon} \text{ is additive noise} \\ H(\mathcal{T}; \boldsymbol{X}\boldsymbol{\epsilon}) = -\sum_{\boldsymbol{Y} \in \mathcal{Y}} p(\boldsymbol{Y}|\boldsymbol{X}\boldsymbol{\epsilon}) \log p(\boldsymbol{Y}|\boldsymbol{X}\boldsymbol{\epsilon}) & \boldsymbol{\epsilon} \text{ is multiplicative noise} \end{cases} \tag{7}$$

While the operation of proactively injecting noise in the latent space can be formulated as:

$$\begin{cases} H_L(\mathcal{T}; \boldsymbol{X} + \boldsymbol{\epsilon}) := H(\boldsymbol{Y}; \boldsymbol{X} + \boldsymbol{\epsilon}) - H(\boldsymbol{X}) & \boldsymbol{\epsilon} \text{ is additive noise} \\ H_L(\mathcal{T}; \boldsymbol{X}\boldsymbol{\epsilon}) := H(\boldsymbol{Y}; \boldsymbol{X}\boldsymbol{\epsilon}) - H(\boldsymbol{X}) & \boldsymbol{\epsilon} \text{ is multiplicative noise} \end{cases} \tag{8}$$

The definition of Eq. 8 differs from the conventional definition, as our method injects the noise into the latent representations instead of the original images. The Gaussian noise is additive, the linear transform noise is also additive, while the salt-and-pepper is multiplicative.

**Gaussian Noise** The Gaussian noise is one of the most common additive noises that appear in computer vision tasks. The Gaussian noise is independent and stochastic, obeying the Gaussian distribution. Without loss of generality, defined as $\mathcal{N}(\mu, \sigma^2)$. Since our injection happens in the latent space, therefore, the complexity of the task is:

$$H_L(\mathcal{T}; \boldsymbol{X} + \mathcal{N}) = H(\boldsymbol{Y}; \boldsymbol{X} + \mathcal{N}) - H(\boldsymbol{X}). \tag{9}$$

We assume that both $\boldsymbol{X}$ and $\boldsymbol{Y}$ follow a multivariate normal distribution, which is a typical assumption in signal processing. Additionally, we can transform the distributions of $\boldsymbol{X}$ and $\boldsymbol{Y}$ to make them (approximately) follow the multivariate normal distribution, even if they initially do not Box & Cox (1964) Feng et al. (2014). According to the definition in Equation 4, the entropy change with

Gaussian noise is:

$$
\begin{aligned}
\triangle S(\mathcal{T}, \mathcal{N}) =& H(\mathcal{T}; \boldsymbol{X}) - H_L(\mathcal{T}; \boldsymbol{X} + \mathcal{N}) \\
=& H(\boldsymbol{Y}; \boldsymbol{X}) - H(\boldsymbol{X}) - (H(\boldsymbol{Y}; \boldsymbol{X} + \mathcal{N}) - H(\boldsymbol{X})) \\
=& H(\boldsymbol{Y}; \boldsymbol{X}) - H(\boldsymbol{Y}; \boldsymbol{X} + \mathcal{N}) \\
=& \frac{1}{2} \log \frac{|\boldsymbol{\Sigma_X}||\boldsymbol{\Sigma_Y} - \boldsymbol{\Sigma_{YX}}\boldsymbol{\Sigma_X^{-1}}\boldsymbol{\Sigma_{XY}}|}{|\boldsymbol{\Sigma_{X+\mathcal{N}}}||\boldsymbol{\Sigma_Y} - \boldsymbol{\Sigma_{YX}}\boldsymbol{\Sigma_{X+\mathcal{N}}^{-1}}\boldsymbol{\Sigma_{XY}}|} \\
=& \frac{1}{2} \log \frac{1}{(1 + \sigma_\epsilon^2 \sum_{i=1}^k \frac{1}{\sigma_{X_i}^2})(1 + \lambda \sum_{i=1}^k \frac{\mathrm{cov}^2(X_i, Y_i)}{\sigma_{X_i}^2(\sigma_{X_i}^2 \sigma_{Y_i}^2 - \mathrm{cov}^2(X_i, Y_i))})}
\end{aligned}
\tag{10}
$$

where $\lambda = \frac{\sigma_\epsilon^2}{1 + \sum_{i=1}^k \frac{1}{\sigma_{X_i}^2}}$, $\sigma_\epsilon^2$ is the variance of the Gaussian noise, $\mathrm{cov}(X_i, Y_i)$ is the covariance of sample pair $X_i, Y_i$, $\sigma_{X_i}^2$ and $\sigma_{Y_i}^2$ are the variance of data sample $X_i$ and data label $Y_i$, respectively. The detailed derivations can be found in the supplementary. Since $\epsilon^2 \geq 0$ and $\lambda \geq 0$, $\sigma_{X_i}^2 \sigma_{Y_i}^2 - \mathrm{cov}^2(X_i, Y_i) = \sigma_{X_i}^2 \sigma_{Y_i}^2 (1 - \rho_{X_i Y_i}^2) \geq 0$, where $\rho_{X_i Y_i}$ is the correlation coefficient, the denominator is greater than 1. Therefore, the entropy change is negative. We can conclude that Gaussian noise injected into the latent space is harmful to the task.

**Linear Transform Noise** This type of noise is obtained by a linear transformation of the features matrix, i.e., $\epsilon = Q\boldsymbol{X}$, where $Q$ is a linear transformation matrix. We name the $Q$ the quality matrix since it controls the property of linear transform noise and determines whether positive or harmful. In the case of linear transform noise injection in the latent space, the task complexity is:

$$
H_L(\mathcal{T}; \boldsymbol{X} + Q\boldsymbol{X}) = H(\boldsymbol{Y}; \boldsymbol{X} + Q\boldsymbol{X}) - H(\boldsymbol{X})
\tag{11}
$$

The entropy change is then formulated as:

$$
\begin{aligned}
\triangle S(\mathcal{T}, Q\boldsymbol{X}) =& H(\mathcal{T}; \boldsymbol{X}) - H_L(\mathcal{T}; \boldsymbol{X} + Q\boldsymbol{X}) \\
=& H(\boldsymbol{Y}; \boldsymbol{X}) - H(\boldsymbol{X}) - (H(\boldsymbol{Y}; \boldsymbol{X} + Q\boldsymbol{X}) - H(\boldsymbol{X})) \\
=& H(\boldsymbol{Y}; \boldsymbol{X}) - H(\boldsymbol{Y}; \boldsymbol{X} + Q\boldsymbol{X}) \\
=& \frac{1}{2} \log \frac{|\boldsymbol{\Sigma_X}||\boldsymbol{\Sigma_Y} - \boldsymbol{\Sigma_{YX}}\boldsymbol{\Sigma_X^{-1}}\boldsymbol{\Sigma_{XY}}|}{|\boldsymbol{\Sigma_{(I+Q)X}}||\boldsymbol{\Sigma_Y} - \boldsymbol{\Sigma_{YX}}\boldsymbol{\Sigma_X^{-1}}\boldsymbol{\Sigma_{XY}}|} \\
=& \frac{1}{2} \log \frac{1}{|I + Q|^2} \\
=& -\log |I + Q|
\end{aligned}
\tag{12}
$$

Since we want the entropy change to be greater than 0, we can formulate Equation 12 as an optimization problem:

$$
\begin{aligned}
\max_Q & \; \triangle S(\mathcal{T}, Q\boldsymbol{X}) \\
s.t. \; & rank(I + Q) = k \\
& Q \sim I \\
& [I + Q]_{ii} \geq [I + Q]_{ij}, i \neq j \\
& \|[I + Q]_i\|_1 = 1
\end{aligned}
\tag{13}
$$

where $\sim$ means the row equivalence. The key to determining whether the linear transform is positive noise or not lies in the matrix of $Q$. The most important step is to ensure that $I + Q$ is reversible, which is $|(I + Q)| \neq 0$. The third constraint is to make the trained classifier get enough information about a specific image and correctly predict the corresponding label. For example, for an image $X_1$ perturbed by another image $X_2$, the classifier obtained dominant information from $X_1$ so that it can predict the label $Y_1$. However, if the perturbing image $X_2$ is dominant, the classifier can hardly predict the correct label $Y_1$ and is more likely to predict as $Y_2$. The fourth constraint is to maintain the norm of latent representations. More in-depth discussion and linear transform noise added to the image level are provided in the supplementary.

**Salt-and-pepper Noise** The salt-and-pepper noise is a common multiplicative noise for images. The image can exhibit unnatural changes, such as black pixels in bright areas or white pixels in dark

areas, specifically as a result of the signal disruption caused by sudden strong interference or bit transmission errors. In the Salt-and-pepper noise case, the entropy change is:

$$
\begin{aligned}
\triangle S(\mathcal{T}, \boldsymbol{\epsilon}) =& H(\mathcal{T}; \boldsymbol{X}) - H_L(\mathcal{T}; \boldsymbol{X}\boldsymbol{\epsilon}) \\
=& H(\boldsymbol{Y}; \boldsymbol{X}) - H(\boldsymbol{X}) - (H(\boldsymbol{Y}; \boldsymbol{X}\boldsymbol{\epsilon}) - H(\boldsymbol{X})) \\
=& H(\boldsymbol{Y}; \boldsymbol{X}) - H(\boldsymbol{Y}; \boldsymbol{X}\boldsymbol{\epsilon}) \\
=& -\sum_{\boldsymbol{X}\in\mathcal{X}}\sum_{\boldsymbol{Y}\in\mathcal{Y}} p(\boldsymbol{X}, \boldsymbol{Y}) \log p(\boldsymbol{X}, \boldsymbol{Y}) + \sum_{\boldsymbol{X}\in\mathcal{X}}\sum_{\boldsymbol{Y}\in\mathcal{Y}}\sum_{\boldsymbol{\epsilon}\in\mathcal{E}} p(\boldsymbol{X}\boldsymbol{\epsilon}, \boldsymbol{Y}) \log p(\boldsymbol{X}\boldsymbol{\epsilon}, \boldsymbol{Y}) \\
=& \mathbb{E}\left[\log\frac{1}{p(\boldsymbol{X}, \boldsymbol{Y})}\right] - \mathbb{E}\left[\log\frac{1}{p(\boldsymbol{X}\boldsymbol{\epsilon}, \boldsymbol{Y})}\right] \\
=& \mathbb{E}\left[\log\frac{1}{p(\boldsymbol{X}, \boldsymbol{Y})}\right] - \mathbb{E}\left[\log\frac{1}{p(\boldsymbol{X}, \boldsymbol{Y})}\right] - \mathbb{E}\left[\log\frac{1}{p(\boldsymbol{\epsilon})}\right] \\
=& -H(\boldsymbol{\epsilon})
\end{aligned}
\tag{14}
$$

Obviously, the entropy change is smaller than 0, which indicates the complexity is increasing when injecting salt-and-pepper noise into the deep model. As can be foreseen, the salt-and-pepper noise is pure detrimental noise. More details and Salt-and-pepper added to the image level are in the supplementary.

## 5 EXPERIMENTS

In this section, we conduct extensive experiments to explore the influence of various types of noises on deep learning models. We employ popular deep learning architectures, including both CNNs and ViTs, and show that the two kinds of deep models can benefit from the positive noise. We employ deep learning models of various scales, including ViT-Tiny (ViT-T), ViT-Small (ViT-S), ViT-Base (ViT-B), and ViT-Large (ViT-L) for Vision Transformers (ViTs), and ResNet-18, ResNet-34, ResNet-50, and ResNet-101 for ResNet architecture. The details of deep models are presented in the supplementary. Without specific instructions, the noise is injected at the last layer of the deep models. Note that for ResNet models, the number of macro layers is 4, and for each macro layer, different scale ResNet models have different micro sublayers. For example, for ResNet-18, the number of macro layers is 4, and for each macro layer, the number of micro sublayers is 2. The noise is injected at the last micro sublayer of the last macro layer for ResNet models. More experimental settings for ResNet and ViT are detailed in the supplementary.

### 5.1 NOISE SETTING

We utilize the standard normal distribution to generate Gaussian noise in our experiments, ensuring that the noise has zero mean and unit variance. Gaussian noise can be expressed as:

$$
\epsilon \sim \mathcal{N}(0, 1)
\tag{15}
$$

For linear transform noise, we use a quality matrix of as:

$$
Q = -\alpha I + \alpha f(I)
\tag{16}
$$

where $I$ is the identity matrix, $\alpha$ represents the linear transform strength and $f$ is a row cyclic shift operation switching row to the next row, for example, in a $3 \times 3$ matrix, $f$ will move Row 1 to Row 2, Row 2 to Row 3, and Row 3 to Row 1. For salt-and-pepper noise, we also use the parameter $\alpha$ to control the probability of the emergence of salt-and-pepper noise, which can be formulated as:

$$
\begin{cases} max(X) & \text{if } p < \alpha/2 \\ min(X) & \text{if } p > 1 - \alpha/2 \end{cases}
\tag{17}
$$

where $p$ is a probability generated by a random seed, $\alpha \in [0, 1]$, and $X$ is the representation of an image.

### 5.2 IMAGE CLASSIFICATION RESULTS

We implement extensive experiments on large-scale datasets such as ImageNet Deng et al. (2009) and small-scale datasets such as TinyImageNet Le & Yang (2015) using ResNets and ViTs.

Table 1: ResNet with different kinds of noise on ImageNet. Vanilla means the vanilla model without noise. Accuracy is shown in percentage. Gaussian noise used here is subjected to standard normal distribution. Linear transform noise used in this table is designed to be positive noise. The difference is shown in the bracket.

| Model | ResNet-18 | ResNet-34 | ResNet-50 | ResNet-101 |
|---|---|---|---|---|
| Vanilla | 63.90 (+0.00) | 66.80 (+0.00) | 70.00 (+0.00) | 70.66 (+0.00) |
| + Gaussian Noise | 62.35 (-1.55) | 65.40 (-1.40) | 69.62 (-0.33) | 70.10 (-0.56) |
| + Linear Transform Noise | **79.62 (+15.72)** | **80.05 (+13.25)** | **81.32 (+11.32)** | **81.91 (+11.25)** |
| + Salt-and-pepper Noise | 55.45 (-8.45) | 63.36 (-3.44) | 45.89 (-24.11) | 52.96 (-17.70) |

Table 2: ViT with different kinds of noise on ImageNet. Vanilla means the vanilla model without injecting noise. Accuracy is shown in percentage. Gaussian noise used here is subjected to standard normal distribution. Linear transform noise used in this table is designed to be positive noise. The difference is shown in the bracket. Note **ViT-L is overfitting on ImageNet** Dosovitskiy et al. (2020) Steiner et al. (2021).

| Model | ViT-T | ViT-S | ViT-B | ViT-L |
|---|---|---|---|---|
| Vanilla | 79.34 (+0.00) | 81.88 (+0.00) | 84.33 (+0.00) | 88.64 (+0.00) |
| + Gaussian Noise | 79.10 (-0.24) | 81.80 (-0.08) | 83.41 (-0.92) | 85.92 (-2.72) |
| + Linear Transform Noise | **80.69 (+1.35)** | **87.27 (+5.39)** | **89.99 (+5.66)** | **88.72 (+0.08)** |
| + Salt-and-pepper Noise | 78.64 (-0.70) | 81.75 (-0.13) | 82.40 (-1.93) | 85.15 (-3.49) |

### 5.2.1 CNN FAMILY

The results of ResNets with different noises on ImageNet are in Table 1. As shown in the table, with the design of linear transform noise to be positive noise (PN), ResNet improves the classification accuracy by a large margin. While the salt-and-pepper, which is theoretically harmful noise (HN), degrades the models. Note we did not utilize data augmentation techniques for ResNet experiments except for normalization. The significant results show that positive noise can effectively improve classification accuracy by reducing task complexity.

### 5.2.2 VIT FAMILY

The results of ViT with different noises on ImageNet are in Table 2. Since the ViT-L is overfitting on the ImageNet Dosovitskiy et al. (2020) Steiner et al. (2021), the positive noise did not work well on the ViT-L. As shown in the table, the existence of positive noise improves the classification accuracy of ViT by a large margin compared to vanilla ViT. The comparisons with previously published works, such as DeiT Touvron et al. (2021), SwinTransformer Liu et al. (2021), DaViT Ding et al. (2022), and MaxViT Tu et al. (2022), are shown in Table 3, and our positive noise-empowered ViT achieved the new state-of-the-art result. Note that the JFT-300M and JFT-4B datasets are private and not publicly available Sun et al. (2017), and we believe that ViT large and above will benefit from positive noise significantly if trained on larger JFT-300M or JFT-4B, which is theoretically supported in section 5.4.

### 5.3 ABLATION STUDY

We also proactively inject noise into variants of ViT, such as DeiT Touvron et al. (2021), Swin Transformer Liu et al. (2021), BEiT Bao et al. (2021), and ConViT d'Ascoli et al. (2021), and the results show that positive noise could benefit various variants of ViT by improving classification accuracy significantly. The results of injecting noise to variants of ViT are reported in the supplementary. We also did ablation studies on the strength of linear transform noise and the injected layer. The results are shown in Fig. 2. We can observe that the deeper layer the positive noise injects, the better prediction performance the model can obtain. There are reasons behind this phenomenon. First, the latent features of input in the deeper layer have better representations than those in shallow layers; second, injection to shallow layers obtain less entropy change gain because of trendy replacing Equation 8 with Equation 7. More results on the small dataset TinyImageNet can be found in the

Table 3: Comparison between Positive Noise Empowered ViT with other ViT variants. Top 1 Accuracy is shown in percentage. Here PN is the positive noise, i.e., linear transform noise.

| Model | Top1 Acc. | Params. | Image Res. | Pretrained Dataset |
|---|---|---|---|---|
| ViT-B Dosovitskiy et al. (2020) | 84.33 | 86M | $224 \times 224$ | ImageNet 21k |
| DeiT-B Touvron et al. (2021) | 85.70 | 86M | $224 \times 224$ | ImageNet 21k |
| SwinTransformer-B Liu et al. (2021) | 86.40 | 88M | $384 \times 384$ | ImageNet 21k |
| MaxViT-B Tu et al. (2022) | 88.82 | 119M | $512 \times 512$ | JFT-300M (Private) |
| ViT-22B Dehghani et al. (2023) | 89.51 | 21743M | $224 \times 224$ | JFT-4B (Private) |
| ViT-B+PN | **89.99** | 86M | $224 \times 224$ | ImageNet 21k |
| ViT-B+PN | **91.37** | 86M | $384 \times 384$ | ImageNet 21k |

Table 4: Top 1 accuracy on ImageNet with the optimal quality matrix of linear transform noise.

| Model | Top1 Acc. | Params. | Image Res. | Pretrained Dataset |
|---|---|---|---|---|
| ViT-B+Optimal Q | **93.87** | 86M | $224 \times 224$ | ImageNet 21k |
| ViT-B+Optimal Q | **95.12** | 86M | $384 \times 384$ | ImageNet 21k |

supplementary. Additionally, we tested the positive linear transformation noise on another popular dataset, the ImageNet V2. The corresponding results are reported in the supplementary.

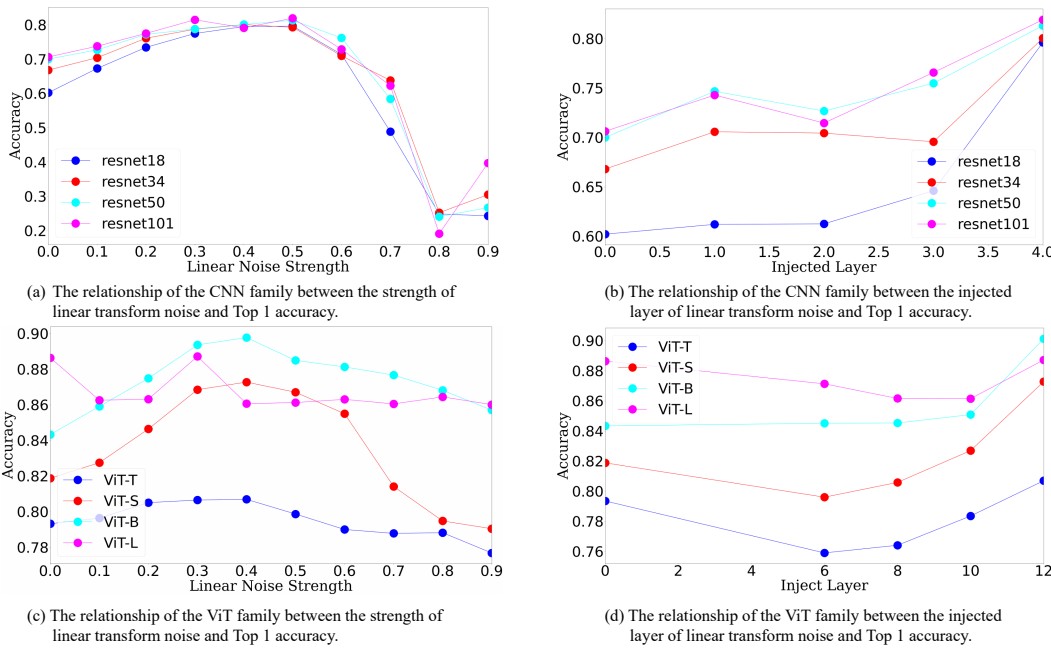

(a) The relationship of the CNN family between the strength of linear transform noise and Top 1 accuracy.

(b) The relationship of the CNN family between the injected layer of linear transform noise and Top 1 accuracy.

(c) The relationship of the ViT family between the strength of linear transform noise and Top 1 accuracy.

(d) The relationship of the ViT family between the injected layer of linear transform noise and Top 1 accuracy.

Figure 2: The relationship between the linear transform noise strength and the top 1 accuracy, and between the injected layer and top 1 accuracy. Parts (a) and (b) are the results of the CNN family, while parts (c) and (d) are the results of the ViT family. For parts (a) and (c) the linear transform noise is injected at the last layer. For parts (b) and (d), the influence of positive noise on different layers is shown. Layers 6, 8, 10, and 12 in the ViT family are chosen for the ablation study.

## 5.4 Optimal Quality Matrix

As shown in Equation 13, it is interesting to learn about the optimal quality matrix of $Q$ that maximizes the entropy change while satisfying the constraints. This equals minimizing the determinant of the matrix sum of $I$ and $Q$. Here, we directly give out the optimal quality matrix of $Q$ as:

$$Q_{optimal} = \text{diag}\left(\frac{1}{k+1} - 1, \ldots, \frac{1}{k+1} - 1\right) + \frac{1}{k+1}\mathbf{1}_{k \times k} \qquad (18)$$

Table 5: Comparison with various ViT-based methods on **Office-Home**.

| Method | Ar→Cl | Ar→Pr | Ar→Re | Cl→Ar | Cl→Pr | Cl→Re | Pr→Ar | Pr→Cl | Pr→Re | Re→Ar | Re→Cl | Re→Pr | Avg. |
|---|---|---|---|---|---|---|---|---|---|---|---|---|---|
| ViT-B | 54.7 | 83.0 | 87.2 | 77.3 | 83.4 | 85.6 | 74.4 | 50.9 | 87.2 | 79.6 | 54.8 | 88.8 | 75.5 |
| TVT-B | 74.9 | 86.8 | 89.5 | 82.8 | 88.0 | 88.3 | 79.8 | 71.9 | 90.1 | 85.5 | 74.6 | 90.6 | 83.6 |
| CDTrans-B | 68.8 | 85.0 | 86.9 | 81.5 | 87.1 | 87.3 | 79.6 | 63.3 | 88.2 | 82.0 | 66.0 | 90.6 | 80.5 |
| SSRT-B | 75.2 | 89.0 | 91.1 | 85.1 | 88.3 | 90.0 | 85.0 | 74.2 | 91.3 | 85.7 | 78.6 | 91.8 | 85.4 |
| TVT-B+PN | **78.3** | **90.6** | **91.9** | **87.8** | **92.1** | **91.9** | **85.8** | **78.7** | **93.0** | **88.6** | **80.6** | **93.5** | **87.7** |

Table 6: Comparison with various ViT-based methods on **Visda2017**.

| Method | plane | bcycl | bus | car | horse | knife | mcycl | person | plant | sktbrd | train | truck | Avg. |
|---|---|---|---|---|---|---|---|---|---|---|---|---|---|
| ViT-B | 97.7 | 48.1 | 86.6 | 61.6 | 78.1 | 63.4 | 94.7 | 10.3 | 87.7 | 47.7 | 94.4 | 35.5 | 67.1 |
| TVT-B | 92.9 | 85.6 | 77.5 | 60.5 | 93.6 | 98.2 | 89.4 | 76.4 | 93.6 | 92.0 | 91.7 | 55.7 | 83.9 |
| CDTrans-B | 97.1 | 90.5 | 82.4 | 77.5 | 96.6 | 96.1 | 93.6 | **88.6** | **97.9** | 86.9 | 90.3 | 62.8 | 88.4 |
| SSRT-B | **98.9** | 87.6 | **89.1** | **84.8** | 98.3 | **98.7** | **96.3** | 81.1 | 94.9 | 97.9 | 94.5 | 43.1 | 88.8 |
| TVT-B+PN | 98.8 | **95.5** | 84.8 | 73.7 | **98.5** | 97.2 | 95.1 | 76.5 | 95.9 | **98.4** | **98.3** | **67.2** | **90.0** |

where $k$ is the number of data samples. And the corresponding upper boundary of the entropy change as:

$$\triangle S(\mathcal{T}, Q_{optimal}\boldsymbol{X}) = (k-1)\log(k+1) \tag{19}$$

The details are provided in the supplementary. We find that the upper boundary of the entropy change of injecting positive noise is determined by the number of data samples, i.e., the scale of the dataset. Therefore, the larger the dataset, the better the effect of injecting positive noise into deep models. With the optimal quality matrix and the top 1 accuracy of ViT-B on ImageNet can be further improved to 95% with careful experimental setting, which is shown in Table 4.

## 5.5 DOMAIN ADAPTION RESULTS

Unsupervised domain adaptation (UDA) aims to learn transferable knowledge across the source and target domains with different distributions Pan & Yang (2009) Wei et al. (2018). Recently, transformer-based methods achieved SOTA results on UDA. Therefore, we evaluate the TVT-B Yang et al. (2023), the first work that adopts Transformer-based architecture for UDA, with the positive noise on widely used UDA benchmarks. Here the positive noise is the linear transform noise identical to that used in the classification task. The positive noise is injected into the last layer of the model, the same as the classification task. The datasets include **Office Home** Venkateswara et al. (2017) and **VisDA2017** Peng et al. (2017). Detailed datasets introduction and experiments training settings are in the supplementary. The results are shown in Table 5 and 6. The ViT-B with positive noise achieves better performance than the existing works. These results show that positive noise can improve model generality and therefore, benefit deep models in domain adaptation tasks.

## 6 CONCLUSION

This study delves into the impact of entropy change on learning systems, achieved by proactively injecting various types of noise into deep models. Our work conducts a comprehensive investigation into the impact of common noise types, such as Gaussian noise, linear transform noise, and salt-and-pepper noise, on deep learning systems. We demonstrate that, under specific conditions, noise can positively affect deep models by reducing task complexity. The experimental results show that injecting positive noise into the latent space significantly enhances the prediction performance of deep models in classification and domain adaptation tasks, leading to new state-of-the-art results on ImageNet. These findings hold broad implications for future research and the potential development of more accurate models for improved real-world applications. Furthermore, the theory behind our method is versatile, and we look forward to exploring entropy change in more deep-learning tasks.

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
