# SUPPLEMENTARY FOR
# EXPLORING THE IMPACT OF INFORMATION ENTROPY CHANGE IN LEARNING SYSTEMS

## 1 THE INFLUENCE OF NOISE ON TASK COMPLEXITY

This section shows the detailed derivations of the conclusion of three kinds of noise on the variations of task entropy. As stated in this paper, the noises can be categorized into additive and multiplicative noise. We list the original definition of task complexity and rewrite task complexity with additive and multiplicative noise, separately.

The original task complexity is formulated as Li (2022):

$$H(\mathcal{T}; \boldsymbol{X}) = - \sum_{\boldsymbol{Y} \in \mathcal{Y}} p(\boldsymbol{Y}|\boldsymbol{X}) \log p(\boldsymbol{Y}|\boldsymbol{X}) \tag{1}$$

The images $\boldsymbol{X}$ in the dataset are supposed to be independent of each other, as are the labels $\boldsymbol{Y}$. However, $\boldsymbol{X}$ and $\boldsymbol{Y}$ are not independent because of the correlation between a data sample $X$ and its corresponding label $Y$, the conditional distribution of $\boldsymbol{Y}$ given $\boldsymbol{X}$ will depend on the joint distribution of $\boldsymbol{X}$ and $\boldsymbol{Y}$. Without knowing the joint distribution of $\boldsymbol{X}$ and $\boldsymbol{Y}$, we can not determine the conditional distribution of $\boldsymbol{Y}$ and $\boldsymbol{X}$. Here, we make some slacks for the distribution of $\boldsymbol{X}$ and $\boldsymbol{Y}$. We can transform the unknown distributions of $\boldsymbol{X}$ and $\boldsymbol{Y}$ to approximately conform to normality by utilizing some techniques, such as Box-Cox transformation, log transform, etc Box & Cox (1964) Feng et al. (2014). After approximate transformation, the distribution of $\boldsymbol{X}$ and $\boldsymbol{Y}$ can be expressed as:

$$\boldsymbol{X} \sim \mathcal{N}(\boldsymbol{\mu_X}, \boldsymbol{\Sigma_X}), \boldsymbol{Y} \sim \mathcal{N}(\boldsymbol{\mu_Y}, \boldsymbol{\Sigma_Y}) \tag{2}$$

where

$$\begin{aligned}
\boldsymbol{\mu_X} &= \mathbb{E}[\boldsymbol{X}] = (\mathbb{E}[X_1], \mathbb{E}[X_2], ..., \mathbb{E}[X_k]])^T \\
\boldsymbol{\mu_Y} &= \mathbb{E}[\boldsymbol{Y}] = (\mathbb{E}[Y_1], \mathbb{E}[Y_2], ..., \mathbb{E}[Y_k]])^T \\
\boldsymbol{\Sigma_X} &= \mathbb{E}[(\boldsymbol{X} - \boldsymbol{\mu_X})(\boldsymbol{X} - \boldsymbol{\mu_X})^T] \\
\boldsymbol{\Sigma_Y} &= \mathbb{E}[(\boldsymbol{Y} - \boldsymbol{\mu_Y})(\boldsymbol{Y} - \boldsymbol{\mu_Y})^T]
\end{aligned} \tag{3}$$

$k$ is the number of samples in the dataset, and $T$ represents the transpose of the matrix.

After transformation, the $\boldsymbol{X}$ and $\boldsymbol{Y}$ are subjected to multivairate normal distribution distribution. Then the conditional distribution of $\boldsymbol{Y}$ given $\boldsymbol{X}$ is also normally distributed Mood (1950) Johnson et al. (1995), which can be formulated as:

$$\boldsymbol{Y}|\boldsymbol{X} \sim \mathcal{N}(\mathbb{E}(\boldsymbol{Y}|\boldsymbol{X} = \boldsymbol{x}), var(\boldsymbol{Y}|\boldsymbol{X} = \boldsymbol{x})) \tag{4}$$

where $\mathbb{E}(\boldsymbol{Y}|\boldsymbol{X} = \boldsymbol{x})$ is the mean of the label set $\boldsymbol{Y}$ given a sample $\boldsymbol{X} = \boldsymbol{x}$ from the dataset, and $var(\boldsymbol{Y}|\boldsymbol{X} = \boldsymbol{x})$ is the variance of $\boldsymbol{Y}$ given a sample from the dataset. The conditional mean $\mathbb{E}[(\boldsymbol{Y}|\boldsymbol{X} = \boldsymbol{x})]$ and conditional variance $var(\boldsymbol{Y}|\boldsymbol{X} = X)$ can be calculated as:

$$\boldsymbol{\mu_{Y|X=x}} = \mathbb{E}[(\boldsymbol{Y}|\boldsymbol{X} = \boldsymbol{x})] = \boldsymbol{\mu_Y} + \boldsymbol{\Sigma_{YX}} \boldsymbol{\Sigma_X^{-1}}(\boldsymbol{x} - \boldsymbol{\mu_X}) \tag{5}$$

$$\boldsymbol{\Sigma_{Y|X=x}} = var(\boldsymbol{Y}|\boldsymbol{X} = \boldsymbol{x}) = \boldsymbol{\Sigma_Y} - \boldsymbol{\Sigma_{YX}} \boldsymbol{\Sigma_X^{-1}} \boldsymbol{\Sigma_{XY}} \tag{6}$$

where $\boldsymbol{\Sigma_{YX}}$ and $\boldsymbol{\Sigma_{XY}}$ are the cross-covariance matrices between $\boldsymbol{Y}$ and $\boldsymbol{X}$, and between $\boldsymbol{X}$ and $\boldsymbol{Y}$, respectively, and $\boldsymbol{\Sigma_X^{-1}}$ denotes the inverse of the covariance matrix of $\boldsymbol{X}$.

Now, let $\boldsymbol{Z} = \boldsymbol{Y}|\boldsymbol{X}$, we shall obtain the task complexity:

$$
\begin{aligned}
H(\mathcal{T}; \boldsymbol{X}) =& - \sum_{\boldsymbol{Y} \in \mathcal{Y}} p(\boldsymbol{Y}|\boldsymbol{X}) \log p(\boldsymbol{Y}|\boldsymbol{X}) \\
=& - \mathbb{E}[\log p(\boldsymbol{Y}|\boldsymbol{X})] \\
=& - \mathbb{E}[\log[(2\pi)^{-k/2}|\boldsymbol{\Sigma}_{\boldsymbol{Z}}|^{-1/2} \exp(-\frac{1}{2}(\boldsymbol{Z} - \boldsymbol{\mu}_{\boldsymbol{Z}})^T \boldsymbol{\Sigma}_{\boldsymbol{Z}}^{-1}(\boldsymbol{Z} - \boldsymbol{\mu}_{\boldsymbol{Z}}))]] \\
=& \frac{k}{2} \log(2\pi) + \frac{1}{2} \log |\boldsymbol{\Sigma}_{\boldsymbol{Z}}| + \frac{1}{2} \mathbb{E}[(\boldsymbol{Z} - \boldsymbol{\mu}_{\boldsymbol{Z}})^T \Sigma_{\boldsymbol{Z}}^{-1}(\boldsymbol{Z} - \boldsymbol{\mu}_{\boldsymbol{Z}})] \\
=& \frac{k}{2}(1 + \log(2\pi)) + \frac{1}{2} \log |\boldsymbol{\Sigma}_{\boldsymbol{Z}}|
\end{aligned}
\tag{7}
$$

where

$$
\begin{aligned}
\mathbb{E}[(\boldsymbol{Z} - \boldsymbol{\mu}_{\boldsymbol{Z}})^T \Sigma_{\boldsymbol{Z}}^{-1}(\boldsymbol{Z} - \boldsymbol{\mu}_{\boldsymbol{Z}})] =& \mathbb{E}[tr((\boldsymbol{Z} - \boldsymbol{\mu}_{\boldsymbol{Z}})^T \Sigma_{\boldsymbol{Z}}^{-1}(\boldsymbol{Z} - \boldsymbol{\mu}_{\boldsymbol{Z}}))] \\
=& \mathbb{E}[tr(\Sigma_{\boldsymbol{Z}}^{-1}(\boldsymbol{Z} - \boldsymbol{\mu}_{\boldsymbol{Z}})(\boldsymbol{Z} - \boldsymbol{\mu}_{\boldsymbol{Z}})^T)] \\
=& tr(\Sigma_{\boldsymbol{Z}}^{-1}(\boldsymbol{Z} - \boldsymbol{\mu}_{\boldsymbol{Z}})(\boldsymbol{Z} - \boldsymbol{\mu}_{\boldsymbol{Z}})^T) \\
=& tr(\Sigma_{\boldsymbol{Z}}^{-1} \Sigma_{\boldsymbol{Z}}) \\
=& tr(\boldsymbol{I}_k) \\
=& k
\end{aligned}
\tag{8}
$$

Therefore, for a specific dataset, we can find that the task entropy is only related to the variance of the $\boldsymbol{Z}$.

However, as we proactively inject additional information into the latent space, the task complexity changes and is defined as :

$$
\begin{cases}
H_L(\mathcal{T}; \boldsymbol{X} + \boldsymbol{\epsilon}) := H(\boldsymbol{Y}; \boldsymbol{X} + \boldsymbol{\epsilon}) - H(\boldsymbol{X}) & \boldsymbol{\epsilon} \text{ is additive noise} \\
H_L(\mathcal{T}; \boldsymbol{X}\boldsymbol{\epsilon}) := H(\boldsymbol{Y}; \boldsymbol{X}\boldsymbol{\epsilon}) - H(\boldsymbol{X}) & \boldsymbol{\epsilon} \text{ is multiplicative noise}
\end{cases}
\tag{9}
$$

Eq. 9 differs from the conventional definition, as our method injects the noise into the latent representations instead of the original images. If adding noise to the original images, then we have the classic definition:

$$
\begin{cases}
H(\mathcal{T}; \boldsymbol{X} + \boldsymbol{\epsilon}) = H(\boldsymbol{Y}; \boldsymbol{X} + \boldsymbol{\epsilon}) - H(\boldsymbol{X} + \boldsymbol{\epsilon}) & \boldsymbol{\epsilon} \text{ is additive noise} \\
H(\mathcal{T}; \boldsymbol{X}\boldsymbol{\epsilon}) = H(\boldsymbol{Y}; \boldsymbol{X}\boldsymbol{\epsilon}) - H(\boldsymbol{X}\boldsymbol{\epsilon}) & \boldsymbol{\epsilon} \text{ is multiplicative noise}
\end{cases}
\tag{10}
$$

## 1.1 Influence of Gaussian Noise on Task Complexity

Gaussian is one of the most common noises in image processing, and it is an additive noise. The Gaussian noise $\boldsymbol{\epsilon}$ is subjected to the normal distribution of $\boldsymbol{\epsilon} \sim \mathcal{N}(\mu_\epsilon, \sigma_\epsilon)$ and is independent of $\boldsymbol{X}$ and $\boldsymbol{Y}$. As we stated the noise can be added to the original images or injected into the latent space, therefore, we discuss the conditions separately.

### 1.1.1 Inject Gaussian Noise in Latent Space

In this case, the task complexity is formulated as:

$$
H_L(\mathcal{T}; \boldsymbol{X} + \boldsymbol{\epsilon}) = H(\boldsymbol{Y}; \boldsymbol{X} + \boldsymbol{\epsilon}) - H(\boldsymbol{X}).
\tag{11}
$$

Take advantage of Eq. 7, thus, the entropy change of injecting Gaussian noise in the latent space can be formulated as:

$$
\begin{aligned}
\triangle S(\mathcal{T}, \boldsymbol{\epsilon}) =& H(\mathcal{T}; \boldsymbol{X}) - H_L(\mathcal{T}; \boldsymbol{X} + \boldsymbol{\epsilon}) \\
=& H(\boldsymbol{Y}; \boldsymbol{X}) - H(\boldsymbol{X}) - (H(\boldsymbol{Y}; \boldsymbol{X} + \boldsymbol{\epsilon}) - H(\boldsymbol{X})) \\
=& H(\boldsymbol{Y}; \boldsymbol{X}) - H(\boldsymbol{Y}; \boldsymbol{X} + \boldsymbol{\epsilon}) \\
=& H(\boldsymbol{Y}|\boldsymbol{X}) + H(\boldsymbol{X}) - (H(\boldsymbol{Y}|\boldsymbol{X} + \boldsymbol{\epsilon}) + H(\boldsymbol{X} + \boldsymbol{\epsilon})) \\
=& \frac{1}{2} \log |\boldsymbol{\Sigma}_{\boldsymbol{Y}|\boldsymbol{X}}| + \frac{1}{2} \log |\boldsymbol{\Sigma}_{\boldsymbol{X}}| - \frac{1}{2} \log |\boldsymbol{\Sigma}_{\boldsymbol{Y}|\boldsymbol{X}+\boldsymbol{\epsilon}}| - \frac{1}{2} \log |\boldsymbol{\Sigma}_{\boldsymbol{X}+\boldsymbol{\epsilon}}| \\
=& \frac{1}{2} \log \frac{|\boldsymbol{\Sigma}_{\boldsymbol{X}}||\boldsymbol{\Sigma}_{\boldsymbol{Y}|\boldsymbol{X}}|}{|\boldsymbol{\Sigma}_{\boldsymbol{X}+\boldsymbol{\epsilon}}||\boldsymbol{\Sigma}_{\boldsymbol{Y}|\boldsymbol{X}+\boldsymbol{\epsilon}}|} \\
=& \frac{1}{2} \log \frac{|\boldsymbol{\Sigma}_{\boldsymbol{X}}||\boldsymbol{\Sigma}_{\boldsymbol{Y}} - \boldsymbol{\Sigma}_{\boldsymbol{YX}} \boldsymbol{\Sigma}_{\boldsymbol{X}}^{-1} \boldsymbol{\Sigma}_{\boldsymbol{XY}}|}{|\boldsymbol{\Sigma}_{\boldsymbol{X}+\boldsymbol{\epsilon}}||\boldsymbol{\Sigma}_{\boldsymbol{Y}} - \boldsymbol{\Sigma}_{\boldsymbol{YX}} \boldsymbol{\Sigma}_{\boldsymbol{X}+\boldsymbol{\epsilon}}^{-1} \boldsymbol{\Sigma}_{\boldsymbol{XY}}|}
\end{aligned}
\tag{12}
$$

where $\boldsymbol{\Sigma}_{\boldsymbol{Y}|\boldsymbol{X}+\boldsymbol{\epsilon}} = \boldsymbol{\Sigma}_{\boldsymbol{Y}} - \boldsymbol{\Sigma}_{\boldsymbol{Y}(\boldsymbol{X}+\boldsymbol{\epsilon})} \boldsymbol{\Sigma}_{\boldsymbol{X}+\boldsymbol{\epsilon}}^{-1} \boldsymbol{\Sigma}_{(\boldsymbol{X}+\boldsymbol{\epsilon})\boldsymbol{Y}}$. Since the Gaussian noise is independent of $\boldsymbol{X}$ and $\boldsymbol{Y}$, we have $\boldsymbol{\Sigma}_{\boldsymbol{Y}(\boldsymbol{X}+\boldsymbol{\epsilon})} = \boldsymbol{\Sigma}_{(\boldsymbol{X}+\boldsymbol{\epsilon})\boldsymbol{Y}} = \boldsymbol{\Sigma}_{\boldsymbol{YX}}$. The corresponding proof is:

$$
\begin{aligned}
\boldsymbol{\Sigma}_{(\boldsymbol{X}+\boldsymbol{\epsilon})\boldsymbol{Y}} =& \mathbb{E}\left[(\boldsymbol{X} + \boldsymbol{\epsilon}) - \mu_{\boldsymbol{X}+\boldsymbol{\epsilon}}\right] \mathbb{E}\left[\boldsymbol{Y} - \mu_{\boldsymbol{Y}}\right] \\
=& \mathbb{E}\left[(\boldsymbol{X} + \boldsymbol{\epsilon})\boldsymbol{Y}\right] - \mu_{\boldsymbol{Y}} \mathbb{E}\left[(\boldsymbol{X} + \boldsymbol{\epsilon})\right] - \mu_{\boldsymbol{X}+\boldsymbol{\epsilon}} \mathbb{E}\left[\boldsymbol{Y}\right] + \mu_{\boldsymbol{Y}} \mu_{\boldsymbol{X}+\boldsymbol{\epsilon}} \\
=& \mathbb{E}\left[(\boldsymbol{X} + \boldsymbol{\epsilon})\boldsymbol{Y}\right] - \mu_{\boldsymbol{Y}} \mathbb{E}\left[(\boldsymbol{X} + \boldsymbol{\epsilon})\right] \\
=& \mathbb{E}\left[\boldsymbol{XY}\right] + \mathbb{E}\left[\boldsymbol{\epsilon Y}\right] - \mu_{\boldsymbol{Y}} \mu_{\boldsymbol{X}} - \mu_{\boldsymbol{Y}} \mu_{\boldsymbol{\epsilon}} \\
=& \mathbb{E}\left[\boldsymbol{XY}\right] - \mu_{\boldsymbol{Y}} \mu_{\boldsymbol{X}} \\
=& \boldsymbol{\Sigma}_{\boldsymbol{XY}}
\end{aligned}
\tag{13}
$$

Obviously,

$$
\begin{cases}
\triangle S(\mathcal{T}, \boldsymbol{\epsilon}) > 0 & if \; \frac{|\boldsymbol{\Sigma}_{\boldsymbol{X}}||\boldsymbol{\Sigma}_{\boldsymbol{Y}|\boldsymbol{X}}|}{|\boldsymbol{\Sigma}_{\boldsymbol{X}+\boldsymbol{\epsilon}}||\boldsymbol{\Sigma}_{\boldsymbol{Y}|\boldsymbol{X}+\boldsymbol{\epsilon}}|} > 1 \\
\triangle S(\mathcal{T}, \boldsymbol{\epsilon}) \leq 0 & if \; \frac{|\boldsymbol{\Sigma}_{\boldsymbol{X}}||\boldsymbol{\Sigma}_{\boldsymbol{Y}|\boldsymbol{X}}|}{|\boldsymbol{\Sigma}_{\boldsymbol{X}+\boldsymbol{\epsilon}}||\boldsymbol{\Sigma}_{\boldsymbol{Y}|\boldsymbol{X}+\boldsymbol{\epsilon}}|} \leq 1
\end{cases}
\tag{14}
$$

To find the relationship between $|\boldsymbol{\Sigma}_{\boldsymbol{X}}||\boldsymbol{\Sigma}_{\boldsymbol{Y}|\boldsymbol{X}}|$ and $|\boldsymbol{\Sigma}_{\boldsymbol{X}+\boldsymbol{\epsilon}}||\boldsymbol{\Sigma}_{\boldsymbol{Y}|\boldsymbol{X}+\boldsymbol{\epsilon}}|$, we need to determine the subterms in each of them. As we mentioned in the previous section, the data samples are independent of each other, and so are the labels.

$$
\begin{aligned}
\boldsymbol{\Sigma}_{\boldsymbol{Y}} =& \mathbb{E}[(\boldsymbol{Y} - \boldsymbol{\mu}_{\boldsymbol{Y}})(\boldsymbol{Y} - \boldsymbol{\mu}_{\boldsymbol{Y}})^T] \\
=& \mathbb{E}[\boldsymbol{Y}\boldsymbol{Y}^T] - \boldsymbol{\mu}_{\boldsymbol{Y}} \boldsymbol{\mu}_{\boldsymbol{Y}}^T \\
=& \mathrm{diag}(\sigma_{Y_1}^2, ..., \sigma_{Y_k}^2)
\end{aligned}
\tag{15}
$$

where

$$
\begin{cases}
\mathbb{E}\left[Y_i Y_j\right] - \mu_{Y_i} \mu_{Y_j} = 0, & i \neq j \\
\mathbb{E}\left[Y_i Y_j\right] - \mu_{Y_i} \mu_{Y_j} = \sigma_{Y_i}^2, & i = j
\end{cases}
\tag{16}
$$

The same procedure can be applied to $\boldsymbol{\Sigma}_{\boldsymbol{Y}(\boldsymbol{X}+\boldsymbol{\epsilon})}$ and $\boldsymbol{\Sigma}_{\boldsymbol{X}+\boldsymbol{\epsilon}}$. Therefore, We can obtain that $\boldsymbol{\Sigma}_{\boldsymbol{Y}} = \mathrm{diag}(\sigma_{Y_1}^2, ..., \sigma_{Y_k}^2)$,

$$
\boldsymbol{\Sigma}_{\boldsymbol{Y}(\boldsymbol{X}+\boldsymbol{\epsilon})} = \mathrm{diag}(\mathrm{cov}(Y_1, X_1 + \epsilon), ..., \mathrm{cov}(Y_k, X_k + \epsilon))
\tag{17}
$$

and $\boldsymbol{\Sigma}_{\boldsymbol{X}+\boldsymbol{\epsilon}}$ is:

$$
\begin{aligned}
\boldsymbol{\Sigma}_{\boldsymbol{X}+\boldsymbol{\epsilon}} =&
\begin{bmatrix}
\sigma_{X_1}^2 + \sigma_\epsilon^2 & \sigma_\epsilon^2 & ... & \sigma_\epsilon^2 & \sigma_\epsilon^2 \\
\sigma_\epsilon^2 & \sigma_{X_2}^2 + \sigma_\epsilon^2 & ... & \sigma_\epsilon^2 & \sigma_\epsilon^2 \\
\vdots & \vdots & & \vdots & \vdots \\
\sigma_\epsilon^2 & \sigma_\epsilon^2 & ... & \sigma_{X_{k-1}}^2 + \sigma_\epsilon^2 & \sigma_\epsilon^2 \\
\sigma_\epsilon^2 & \sigma_\epsilon^2 & ... & \sigma_\epsilon^2 & \sigma_{X_k}^2 + \sigma_\epsilon^2
\end{bmatrix} \\
=& \mathrm{diag}(\sigma_{X_1}^2, ..., \sigma_{X_k}^2)\boldsymbol{I}_k + \sigma_\epsilon^2 \mathbf{1}_k
\end{aligned}
\tag{18}
$$

where $\boldsymbol{I}_k$ is a $k \times k$ identity matrix and $\boldsymbol{1}_k$ is a all ones $k \times k$ matrix. We use $\boldsymbol{U}$ to represent $\mathrm{diag}(\sigma^2_{X_1}, ..., \sigma^2_{X_k})\boldsymbol{I}_k$, and $\boldsymbol{u}$ to represent a all ones vector $[1, ..., 1]^T$. Thanks to the Sherman–Morrison Formula Sherman (1949) and Woodbury Formula Woodbury (1950), we can obtain the inverse of $\boldsymbol{\Sigma_{X+\epsilon}}$ as:

$$
\begin{aligned}
\boldsymbol{\Sigma^{-1}_{X+\epsilon}} &= (\boldsymbol{U} + \sigma^2_\epsilon \boldsymbol{u}\boldsymbol{u}^T)^{-1} \\
&= \boldsymbol{U}^{-1} - \frac{\sigma^2_\epsilon}{1 + \sigma^2_\epsilon \boldsymbol{u}^T \boldsymbol{U}^{-1} \boldsymbol{u}} \boldsymbol{U}^{-1} \boldsymbol{u}\boldsymbol{u}^T \boldsymbol{U}^{-1} \\
&= \boldsymbol{U}^{-1} - \frac{\sigma^2_\epsilon}{1 + \sum_{i=1}^k \frac{1}{\sigma^2_{X_i}}} \boldsymbol{U}^{-1} \boldsymbol{1}_k \boldsymbol{U}^{-1} \\
&= \lambda \begin{bmatrix}
\frac{1}{\lambda \sigma^2_{X_1}} - \frac{1}{\sigma^4_{X_1}} & -\frac{1}{\sigma^2_{X_1}\sigma^2_{X_2}} & \cdots & -\frac{1}{\sigma^2_{X_1}\sigma^2_{X_{k-1}}} & -\frac{1}{\sigma^2_{X_1}\sigma^2_{X_k}} \\
-\frac{1}{\sigma^2_{X_2}\sigma^2_{X_1}} & \frac{1}{\lambda \sigma^2_{X_2}} - \frac{1}{\sigma^4_{X_2}} & \cdots & -\frac{1}{\sigma^2_{X_2}\sigma^2_{X_{k-1}}} & -\frac{1}{\sigma^2_{X_2}\sigma^2_{X_k}} \\
\vdots & \vdots & & \vdots & \vdots \\
-\frac{1}{\sigma^2_{X_{k-1}}\sigma^2_{X_1}} & -\frac{1}{\sigma^2_{X_{k-1}}\sigma^2_{X_2}} & \cdots & \frac{1}{\lambda \sigma^2_{X_{k-1}}} - \frac{1}{\sigma^4_{X_{k-1}}} & -\frac{1}{\sigma^2_{X_{k-1}}\sigma^2_{X_k}} \\
-\frac{1}{\sigma^2_{X_k}\sigma^2_{X_1}} & -\frac{1}{\sigma^2_{X_k}\sigma^2_{X_2}} & \cdots & -\frac{1}{\sigma^2_{X_k}\sigma^2_{X_{k-1}}} & \frac{1}{\lambda \sigma^2_{X_k}} - \frac{1}{\sigma^4_{X_k}}
\end{bmatrix}
\end{aligned} \tag{19}
$$

where $\boldsymbol{U}^{-1} = \mathrm{diag}((\sigma^2_{X_1})^{-1}, ..., (\sigma^2_{X_k})^{-1})$ and $\lambda = \frac{\sigma^2_\epsilon}{1 + \sum_{i=1}^k \frac{1}{\sigma^2_{X_i}}}$.

Therefore, substitute Equation 19 into $|\boldsymbol{\Sigma_Y} - \boldsymbol{\Sigma_{Y(X+\epsilon)}}\boldsymbol{\Sigma^{-1}_{X+\epsilon}}\boldsymbol{\Sigma_{(X+\epsilon)Y}}|$, we can obtain:

$$
\begin{aligned}
&|\boldsymbol{\Sigma_Y} - \boldsymbol{\Sigma_{Y(X+\epsilon)}}\boldsymbol{\Sigma^{-1}_{X+\epsilon}}\boldsymbol{\Sigma_{(X+\epsilon)Y}}| \\
&= \left| \begin{bmatrix} \sigma^2_{Y_1} & \cdots & 0 \\ \vdots & \ddots & \vdots \\ 0 & \cdots & \sigma^2_{Y_k} \end{bmatrix} - \begin{bmatrix} \mathrm{cov}(Y_1, X_1+\epsilon) & \cdots & 0 \\ \vdots & \ddots & \vdots \\ 0 & \cdots & \mathrm{cov}(Y_k, X_k+\epsilon) \end{bmatrix} \boldsymbol{\Sigma^{-1}_{X+\epsilon}} \begin{bmatrix} \mathrm{cov}(Y_1, X_1+\epsilon) & \cdots & 0 \\ \vdots & \ddots & \vdots \\ 0 & \cdots & \mathrm{cov}(Y_k, X_k+\epsilon) \end{bmatrix} \right| \\
&= \left| \begin{bmatrix} \sigma^2_{Y_1} - \mathrm{cov}^2(Y_1, X_1+\epsilon)(\frac{1}{\sigma^2_{X_1}} - \frac{\lambda}{\sigma^4_{X_1}}) & \cdots & \mathrm{cov}(Y_1, X_1+\epsilon)\mathrm{cov}(Y_k, X_k+\epsilon)\frac{\lambda}{\sigma^2_{X_1}\sigma^2_{X_k}} \\ \vdots & & \vdots \\ \mathrm{cov}(Y_k, X_k+\epsilon)\mathrm{cov}(Y_1, X_1+\epsilon)\frac{\lambda}{\sigma^2_{X_k}\sigma^2_{X_1}} & \cdots & \sigma^2_{Y_k} - \mathrm{cov}^2(Y_k, X_k+\epsilon)(\frac{1}{\sigma^2_{X_k}} - \frac{\lambda}{\sigma^4_{X_k}}) \end{bmatrix} \right| \\
&= \left| \begin{bmatrix} \sigma^2_{Y_1} - \frac{1}{\sigma^2_{X_1}}\mathrm{cov}^2(Y_1, X_1) & & \\ & \ddots & \\ & & \sigma^2_{Y_k} - \frac{1}{\sigma^2_{X_k}}\mathrm{cov}^2(Y_k, X_k) \end{bmatrix} + \lambda \begin{bmatrix} \frac{1}{\sigma^4_{X_1}}\mathrm{cov}^2(Y_1, X_1) & \cdots & \frac{1}{\sigma^2_{X_1}\sigma^2_{X_k}}\mathrm{cov}(Y_1, X_1)\mathrm{cov}(Y_k, X_k) \\ \vdots & & \vdots \\ \frac{1}{\sigma^2_{X_k}\sigma^2_{X_1}}\mathrm{cov}(Y_k, X_k)\mathrm{cov}(Y_1, X_1) & \cdots & \frac{1}{\sigma^4_{X_k}}\mathrm{cov}^2(Y_k, X_k) \end{bmatrix} \right|
\end{aligned} \tag{20}
$$

We use the notation $\boldsymbol{v} = \left[ \frac{1}{\sigma^2_{X_1}}\mathrm{cov}(Y_1, X_1) \quad \cdots \quad \frac{1}{\sigma^2_{X_k}}\mathrm{cov}(Y_k, X_k) \right]^T$, and $\boldsymbol{V} = \mathrm{diag}(\frac{1}{\sigma^2_{X_1}}\mathrm{cov}^2(Y_1, X_1), \cdots, \frac{1}{\sigma^2_{X_k}}\mathrm{cov}^2(Y_k, X_k))$. And utilize the rule of determinants of sums Marcus (1990), then we have:

$$
\begin{aligned}
|\boldsymbol{\Sigma_Y} - \boldsymbol{\Sigma_{Y(X+\epsilon)}}\boldsymbol{\Sigma^{-1}_{X+\epsilon}}\boldsymbol{\Sigma_{(X+\epsilon)Y}}| &= |(\boldsymbol{\Sigma_Y} - \boldsymbol{V}) + \lambda \boldsymbol{v}\boldsymbol{v}^T| \\
&= |\boldsymbol{\Sigma_Y} - \boldsymbol{V}| + \lambda \boldsymbol{v}^T (\boldsymbol{\Sigma_Y} - \boldsymbol{V})^* \boldsymbol{v}
\end{aligned} \tag{21}
$$

where $(\boldsymbol{\Sigma_Y} - \boldsymbol{V})^*$ is the adjoint of the matrix $(\boldsymbol{\Sigma_Y} - \boldsymbol{V})$. For simplicity, we can rewrite $|\boldsymbol{\Sigma_Y} - \boldsymbol{\Sigma_{Y(X+\epsilon)}}\boldsymbol{\Sigma^{-1}_{X+\epsilon}}\boldsymbol{\Sigma_{(X+\epsilon)Y}}|$ as:

$$
\begin{aligned}
&|\boldsymbol{\Sigma_Y} - \boldsymbol{\Sigma_{Y(X+\epsilon)}}\boldsymbol{\Sigma^{-1}_{X+\epsilon}}\boldsymbol{\Sigma_{(X+\epsilon)Y}}| \\
&= \prod_{i=1}^k (\sigma^2_{Y_i} - \mathrm{cov}^2(Y_i, X_i)\frac{1}{\sigma^2_{X_i}}) + \Omega
\end{aligned} \tag{22}
$$

where $\Omega = \lambda \boldsymbol{v}^T (\boldsymbol{\Sigma_Y} - \boldsymbol{V})^* \boldsymbol{v}$. The specific value of $\Omega$ can be obtained as:

$$
\Omega = \lambda \left[ \frac{1}{\sigma^2_{X_1}}\mathrm{cov}(Y_1, X_1) \quad \cdots \quad \frac{1}{\sigma^2_{X_k}}\mathrm{cov}(Y_k, X_k) \right] \begin{bmatrix} V_{11} & & \\ & \ddots & \\ & & V_{kk} \end{bmatrix} \begin{bmatrix} \frac{1}{\sigma^2_{X_1}}\mathrm{cov}(Y_1, X_1) \\ \vdots \\ \frac{1}{\sigma^2_{X_k}}\mathrm{cov}(Y_k, X_k) \end{bmatrix} \tag{23}
$$

where the elements $V_{ii}, i \in [1, k]$ are minors of the matrix and expressed as:

$$V_{ii} = \prod_{j=1, j \neq i}^{k} \left[ \sigma_{Y_j}^2 - \frac{1}{\sigma_{X_j}^2} \text{cov}^2(X_j, Y_j) \right] \tag{24}$$

After some necessary steps, Equation 23 is reduced to:

$$\begin{aligned}
\Omega =& \lambda \sum_{i=1}^{k} \frac{\frac{1}{\sigma_{X_i}^4} \text{cov}^2(Y_i, X_i) \prod_{j=1}^{k}(\sigma_{Y_j}^2 - \text{cov}^2(Y_j, X_j)\frac{1}{\sigma_{X_j}^2})}{(\sigma_{Y_i}^2 - \text{cov}^2(Y_i, X_i)\frac{1}{\sigma_{X_i}^2})} \\
=& \lambda \prod_{i=1}^{k}(\sigma_{Y_i}^2 - \text{cov}^2(Y_i, X_i)\frac{1}{\sigma_{X_i}^2}) \cdot \sum_{i=1}^{k} \frac{\text{cov}^2(X_i, Y_i)}{\sigma_{X_i}^2(\sigma_{X_i}^2 \sigma_{Y_i}^2 - \text{cov}^2(X_i, Y_i))}
\end{aligned} \tag{25}$$

Substitute Equation 25 into Equation 22, we can get:

$$\begin{aligned}
& |\Sigma_Y - \Sigma_{Y(X+\epsilon)} \Sigma_{X+\epsilon}^{-1} \Sigma_{(X+\epsilon)Y}| \\
& = \prod_{i=1}^{k}(\sigma_{Y_i}^2 - \text{cov}^2(Y_i, X_i)\frac{1}{\sigma_{X_i}^2}) \cdot (1 + \lambda \sum_{i=1}^{k} \frac{\text{cov}^2(X_i, Y_i)}{\sigma_{X_i}^2(\sigma_{X_i}^2 \sigma_{Y_i}^2 - \text{cov}^2(X_i, Y_i))})
\end{aligned} \tag{26}$$

Accordingly, $|\Sigma_Y - \Sigma_{YX} \Sigma_X^{-1} \Sigma_{XY}|$ is:

$$|\Sigma_Y - \Sigma_{YX} \Sigma_X^{-1} \Sigma_{XY}| = \prod_{i=1}^{k}(\sigma_{Y_i}^2 - \frac{1}{\sigma_{X_i}^2} \text{cov}^2(X_i, Y_i)) \tag{27}$$

As a result, $\frac{|\Sigma_{Y|X+\epsilon}|}{|\Sigma_{Y|X}|}$ is expressed as:

$$\frac{|\Sigma_{Y|X}|}{|\Sigma_{Y|X+\epsilon}|} = \frac{\prod_{i=1}^{k}(\sigma_{Y_i}^2 - \frac{1}{\sigma_{X_i}^2} \text{cov}^2(X_i, Y_i))}{\prod_{i=1}^{k}(\sigma_{Y_i}^2 - \text{cov}^2(Y_i, X_i)\frac{1}{\sigma_{X_i}^2}) \cdot (1 + \lambda \sum_{i=1}^{k} \frac{\text{cov}^2(X_i, Y_i)}{\sigma_{X_i}^2(\sigma_{X_i}^2 \sigma_{Y_i}^2 - \text{cov}^2(X_i, Y_i))})} \tag{28}$$

Combine Equations 28 and 18 together, the entropy change is expressed as:

$$\triangle S(\mathcal{T}, \epsilon) = \frac{1}{2} \log \frac{1}{(1 + \sigma_\epsilon^2 \sum_{i=1}^{k} \frac{1}{\sigma_{X_i}^2})(1 + \lambda \sum_{i=1}^{k} \frac{\text{cov}^2(X_i, Y_i)}{\sigma_{X_i}^2(\sigma_{X_i}^2 \sigma_{Y_i}^2 - \text{cov}^2(X_i, Y_i))})} \tag{29}$$

It is difficult to tell that Equation 29 is greater or smaller than 1 directly. But one thing for sure is that when there is no Gaussian noise, Equation 29 equals 1. However, we can use another way to compare the numerator and denominator of Equation 29. Instead, we compare the numerator and denominator using subtraction. Let:

$$f(\sigma_\epsilon^2) = 1 - (1 + \sigma_\epsilon^2 \sum_{i=1}^{k} \frac{1}{\sigma_{X_i}^2})(1 + \lambda \sum_{i=1}^{k} \frac{\text{cov}^2(X_i, Y_i)}{\sigma_{X_i}^2(\sigma_{X_i}^2 \sigma_{Y_i}^2 - \text{cov}^2(X_i, Y_i))}) \tag{30}$$

Obviously, the variance $\sigma_\epsilon^2$ of the Gaussian noise control the result of $f(\sigma_\epsilon)$, while the mean $\mu_\epsilon$ has no influence. When $\sigma_\epsilon$ approaching 0, we have:

$$\lim_{\sigma_\epsilon^2 \to 0} f(\sigma_\epsilon^2) = 0 \tag{31}$$

To determine if Gaussian noise can be positive noise, we need to determine whether the entropy change is large or smaller than 0.

$$\begin{cases} \triangle S(\mathcal{T}, \epsilon) > 0 & \text{if } f(\sigma_\epsilon^2) > 0 \\ \triangle S(\mathcal{T}, \epsilon) \leq 0 & \text{if } f(\sigma_\epsilon^2) \leq 0 \end{cases} \tag{32}$$

From the above equations, the sign of the entropy change is determined by the statistical properties of the data samples and labels. Since $\epsilon^2 \geq 0$, $\lambda \geq 0$ and $\sum_{i=1}^{k} \frac{1}{\sigma_{X_i}^2} \geq 0$, we need to have a deep dive into the residual part, i.e.,

$$\sum_{i=1}^{k} \frac{\text{cov}^2(X_i, Y_i)}{\sigma_{X_i}^2(\sigma_{X_i}^2 \sigma_{Y_i}^2 - \text{cov}^2(X_i, Y_i))} = \sum_{i=1}^{k} \frac{\text{cov}^2(X_i, Y_i)}{\sigma_{X_i}^4 \sigma_{Y_i}^2(1 - \rho_{X_i Y_i}^2)} \tag{33}$$

where $\rho_{X_i Y_i}$ is the correlation coefficient, and $\rho_{X_i Y_i}^2 \in [0, 1]$. Eq. 33 is greater than 0, As a result, the sign of the entropy change in the Gaussian noise case is negative. We can conclude that Gaussian noise added to the latent space is harmful to the task.

### 1.1.2 ADD GAUSSIAN NOISE TO ORIGINAL IMAGES

The variation of task complexity by adding Gaussian noise to input images can be formulated as:

$$
\begin{aligned}
\triangle S(\mathcal{T}, \boldsymbol{\epsilon}) =& H(\mathcal{T}; \boldsymbol{X}) - H(\mathcal{T}; \boldsymbol{X} + \boldsymbol{\epsilon}) \\
=& \frac{1}{2} \log |\boldsymbol{\Sigma}_{\boldsymbol{Y}|\boldsymbol{X}}| - \frac{1}{2} \log |\boldsymbol{\Sigma}_{\boldsymbol{Y}|\boldsymbol{X}+\epsilon}| \\
=& \frac{1}{2} \log \frac{|\boldsymbol{\Sigma}_{\boldsymbol{Y}|\boldsymbol{X}}|}{|\boldsymbol{\Sigma}_{\boldsymbol{Y}|\boldsymbol{X}+\epsilon}|} \\
=& \frac{1}{2} \log \frac{|\boldsymbol{\Sigma}_{\boldsymbol{Y}} - \boldsymbol{\Sigma}_{\boldsymbol{Y}\boldsymbol{X}}\boldsymbol{\Sigma}_{\boldsymbol{X}}^{-1}\boldsymbol{\Sigma}_{\boldsymbol{X}\boldsymbol{Y}}|}{|\boldsymbol{\Sigma}_{\boldsymbol{Y}} - \boldsymbol{\Sigma}_{\boldsymbol{Y}(\boldsymbol{X}+\epsilon)}\boldsymbol{\Sigma}_{\boldsymbol{X}+\epsilon}^{-1}\boldsymbol{\Sigma}_{(\boldsymbol{X}+\epsilon)\boldsymbol{Y}}|} \\
=& \frac{1}{2} \log \frac{|\boldsymbol{\Sigma}_{\boldsymbol{Y}} - \boldsymbol{\Sigma}_{\boldsymbol{Y}\boldsymbol{X}}\boldsymbol{\Sigma}_{\boldsymbol{X}}^{-1}\boldsymbol{\Sigma}_{\boldsymbol{X}\boldsymbol{Y}}|}{|\boldsymbol{\Sigma}_{\boldsymbol{Y}} - \boldsymbol{\Sigma}_{\boldsymbol{Y}\boldsymbol{X}}\boldsymbol{\Sigma}_{\boldsymbol{X}+\epsilon}^{-1}\boldsymbol{\Sigma}_{\boldsymbol{X}\boldsymbol{Y}}|}
\end{aligned}
\tag{34}
$$

Borrow the equations from the case of Gaussian noise added to the latent space, we have:

$$
\triangle S(\mathcal{T}, \boldsymbol{\epsilon}) = \frac{1}{2} \log \frac{1}{1 + \lambda \sum_{i=1}^{k} \frac{\mathrm{cov}^2(X_i, Y_i)}{\sigma_{X_i}^2(\sigma_{X_i}^2 \sigma_{Y_i}^2 - \mathrm{cov}^2(X_i, Y_i))}}
\tag{35}
$$

Clearly, the introduction of Gaussian noise to each pixel in the original images has a detrimental impact on the task. **Note** that some studies have empirically shown that adding Gaussian noise to partial pixels of input images may be beneficial to the learning task Li (2022) Zhang et al. (2023).

## 1.2 INFLUENCE OF LINEAR TRANSFORM NOISE ON TASK COMPLEXITY

In our work, the linear transform noise refers to an image or the latent representation of an image that is perturbed by the combination of other images or latent representations of other images.

### 1.2.1 INJECT LINEAR TRANSFORM NOISE IN LATENT SPACE

The entropy change of injecting linear transform noise can be formulated as:

$$
\begin{aligned}
\triangle S(\mathcal{T}, Q\boldsymbol{X}) =& H(\mathcal{T}; \boldsymbol{X}) - H_L(\mathcal{T}; \boldsymbol{X} + Q\boldsymbol{X}) \\
=& H(\boldsymbol{Y}; \boldsymbol{X}) - H(\boldsymbol{X}) - (H(\boldsymbol{Y}; \boldsymbol{X} + Q\boldsymbol{X}) - H(\boldsymbol{X})) \\
=& H(\boldsymbol{Y}; \boldsymbol{X}) - H(\boldsymbol{Y}; \boldsymbol{X} + Q\boldsymbol{X}) \\
=& \frac{1}{2} \log \frac{|\boldsymbol{\Sigma}_{\boldsymbol{X}}||\boldsymbol{\Sigma}_{\boldsymbol{Y}} - \boldsymbol{\Sigma}_{\boldsymbol{Y}\boldsymbol{X}}\boldsymbol{\Sigma}_{\boldsymbol{X}}^{-1}\boldsymbol{\Sigma}_{\boldsymbol{X}\boldsymbol{Y}}|}{|\boldsymbol{\Sigma}_{(I+Q)\boldsymbol{X}}||\boldsymbol{\Sigma}_{\boldsymbol{Y}} - \boldsymbol{\Sigma}_{\boldsymbol{Y}\boldsymbol{X}}\boldsymbol{\Sigma}_{\boldsymbol{X}}^{-1}\boldsymbol{\Sigma}_{\boldsymbol{X}\boldsymbol{Y}}|} \\
=& \frac{1}{2} \log \frac{1}{|I+Q|^2} \\
=& - \log |I+Q|
\end{aligned}
\tag{36}
$$

Since we want the entropy change to be greater than 0, we can formulate Equation 36 as an optimization problem:

$$
\begin{aligned}
\max_{Q} & \triangle S(\mathcal{T}, Q\boldsymbol{X}) \\
s.t. \; & rank(I+Q) = k \\
& Q \sim I \\
& [I+Q]_{ii} \geq [I+Q]_{ij}, i \neq j \\
& \|[I+Q]_i\|_1 = 1
\end{aligned}
\tag{37}
$$

where $\sim$ means the row equivalence. The key to determining whether the linear transform is positive noise or not lies in the matrix of $Q$. The most important step is to ensure that $I+Q$ is reversible, which is $|(I+Q)| \neq 0$. For this, we need to investigate what leads $I+Q$ to be rank-deficient. The third constraint is to make the trained classifier get enough information about a specific image and correctly predict the corresponding label. For example, for an image $X_1$ perturbed by another image

$X_2$, the classifier obtained dominant information from $X_1$ so that it can predict the label $Y_1$. However, if the perturbed image $X_2$ is dominant, the classifier can hardly predict the correct label $Y_1$. The fourth constraint is the normalization of latent representations.

**Rank Deficiency Cases** To avoid causing a rank deficiency of $I + Q$, we need to figure out the conditions that lead to rank deficiency. Here we show a simple case causing the rank deficiency. When the matrix $Q$ is a backward identity matrix Horn & R. (2012),

$$Q_{i,j} = \begin{cases} 1, & i + j = k + 1 \\ 0, & i + j \neq k + 1 \end{cases} \tag{38}$$

i.e.,

$$Q = \begin{bmatrix} 0 & 0 & ... & 0 & 0 & 1 \\ 0 & 0 & ... & 0 & 1 & 0 \\ \vdots & \vdots & & \vdots & \vdots & \vdots \\ 0 & 1 & ... & 0 & 0 & 0 \\ 1 & 0 & ... & 0 & 0 & 0 \end{bmatrix} \tag{39}$$

then $(I + Q)$ will be:

$$I + Q = \begin{bmatrix} 1 & 0 & ... & 0 & 0 & 1 \\ 0 & 1 & ... & 0 & 1 & 0 \\ \vdots & \vdots & & \vdots & \vdots & \vdots \\ 0 & 1 & ... & 0 & 1 & 0 \\ 1 & 0 & ... & 0 & 0 & 1 \end{bmatrix} \tag{40}$$

Thus, $I + Q$ will be rank-deficient when $Q$ is a backward identity. In fact, when the following constraints are satisfied, the $I + Q$ will be rank-deficient:

$$\text{HermiteForm}(I + Q)_i = \mathbf{0}, \quad \exists i \in [1, k] \tag{41}$$

where index $i$ is the row index, in this paper, the row index starts from 1, and $\text{HermiteForm}$ is the Hermite normal form Kannan & Bachem (1979).

**Full Rank Cases** Except for the rank deficiency cases, $I + Q$ has full rank and is reversible. Since $Q$ is a row equivalent to the identity matrix, we need to introduce the three types of elementary row operations as follows Shores (2007).

▷ 1 **Row Swap** Exchange rows.
Row swap here allows exchanging any number of rows. This is slightly different from the original one that only allows any two rows exchange since following the original row swap will lead to a rank deficiency. When the $Q$ is derived from $I$ with **Row Swap**, it will break the third constraint. Therefore, **Row Swap** merely is considered harmful and would degrade the deep model.

▷ 2 **Scalar Multiplication** Multiply any row by a constant $\beta$. This breaks the fourth constraint, thus degrading the deep models.

▷ 3 **Row Sum** Add a multiple of one row to another row. Then the matrix $I + Q$ would be like:

$$I + Q = \begin{bmatrix} 1 & & & & \\ & \cdot & & & \\ & & \cdot & & \\ & & & \cdot & \\ & & & & 1 \end{bmatrix} + \begin{bmatrix} 1 & & & & \\ & \cdot & & \beta & \\ & & \cdot & & \\ & & & \cdot & \\ & & & & 1 \end{bmatrix}$$

$$= \begin{bmatrix} 2 & & & & \\ & \cdot & & \beta & \\ & & \cdot & & \\ & & & \cdot & \\ & & & & 2 \end{bmatrix} \tag{42}$$

where $\beta$ can be at a random position beside the diagonal. As we can see from the simple example, **Row Sum** breaks the fourth constraint and makes entropy change smaller than 0.

From the above discussion, none of the single elementary row operations can guarantee positive effects on deep models.

However, if we combine the elementary row operations, it is possible to make the entropy change greater than 0 as well as satisfy the constraints. For example, we combine the **Row Swap** and **Scalar Multiplication** to generate the $Q$:

$$
\begin{aligned}
I + Q &= \begin{bmatrix} 1 & & & \\ & \cdot & & \\ & & \cdot & \\ & & & \cdot \\ & & & & 1 \end{bmatrix} + \begin{bmatrix} -0.5 & 0.5 & & & \\ & \cdot & \cdot & & \\ & & \cdot & \cdot & \\ & & & \cdot & 0.5 \\ 0.5 & & & & -0.5 \end{bmatrix} \\
&= \begin{bmatrix} 0.5 & 0.5 & & & \\ & \cdot & \cdot & & \\ & & \cdot & \cdot & \\ & & & \cdot & 0.5 \\ 0.5 & & & & 0.5 \end{bmatrix}
\end{aligned}
\tag{43}
$$

In this case, $\triangle S(\mathcal{T}, Q\boldsymbol{X}) > 0$ when $Q = -0.5I$. The constraints are satisfied. This is just a simple case of adding linear transform noise that benefits deep models. Actually, there exists a design space of $Q$ that within the design space, deep models can reduce task entropy by injecting linear transform noise. To this end, we demonstrate that linear transform can be positive noise.

### 1.2.2 ADD LINEAR TRANSFORM NOISE TO ORIGINAL IMAGES

The task entropy with linear transform noise can be formulated as:

$$
\begin{aligned}
H(\mathcal{T}; \boldsymbol{X} + Q\boldsymbol{X}) &= -\sum_{\boldsymbol{Y} \in \mathcal{Y}} p(\boldsymbol{Y}|\boldsymbol{X} + Q\boldsymbol{X}) \log p(\boldsymbol{Y}|\boldsymbol{X} + Q\boldsymbol{X}) \\
&= -\sum_{\boldsymbol{Y} \in \mathcal{Y}} p(\boldsymbol{Y}|(I + Q)\boldsymbol{X}) \log p(\boldsymbol{Y}|(I + Q)\boldsymbol{X})
\end{aligned}
\tag{44}
$$

where $I$ is an identity matrix, and $Q$ is derived from $I$ using elementary row operations. The conditional distribution of $\boldsymbol{Y}$ given $\boldsymbol{X} + Q\boldsymbol{X}$ is also multivariate subjected to the normal distribution, which can be formulated as:

$$
\boldsymbol{Y}|(I + Q)\boldsymbol{X} \sim \mathcal{N}(\mathbb{E}(\boldsymbol{Y}|(I + Q)\boldsymbol{X}), var(\boldsymbol{Y}|(I + Q)\boldsymbol{X}))
\tag{45}
$$

The linear transform on $\boldsymbol{X}$ does not change the distribution of the $\boldsymbol{X}$. It is not difficult to obtain:

$$
\boldsymbol{\mu}_{\boldsymbol{Y}|(I+Q)\boldsymbol{X}} = \boldsymbol{\mu}_{\boldsymbol{Y}} + \boldsymbol{\Sigma}_{\boldsymbol{YX}}\boldsymbol{\Sigma}_{\boldsymbol{X}}^{-1}(I + Q)^{-1}((I + Q)X - (I + Q)\boldsymbol{\mu}_{\boldsymbol{X}})
\tag{46}
$$

$$
\boldsymbol{\Sigma}_{(\boldsymbol{Y}|(I+Q)\boldsymbol{X})} = \boldsymbol{\Sigma}_{\boldsymbol{Y}} - \boldsymbol{\Sigma}_{\boldsymbol{YX}}\boldsymbol{\Sigma}_{\boldsymbol{X}}^{-1}\boldsymbol{\Sigma}_{\boldsymbol{XY}}
\tag{47}
$$

Thus, the variation of task entropy adding linear transform noise can be formulated as:

$$
\begin{aligned}
\triangle S(\mathcal{T}, Q\boldsymbol{X}) &= H(\mathcal{T}; \boldsymbol{X}) - H(\mathcal{T}; \boldsymbol{X} + Q\boldsymbol{X}) \\
&= \frac{1}{2} \log |\boldsymbol{\Sigma}_{\boldsymbol{Y}|\boldsymbol{X}}| - \frac{1}{2} \log |\boldsymbol{\Sigma}_{\boldsymbol{Y}|\boldsymbol{X}+Q\boldsymbol{X}}| \\
&= \frac{1}{2} \log \frac{|\boldsymbol{\Sigma}_{\boldsymbol{Y}|\boldsymbol{X}}|}{|\boldsymbol{\Sigma}_{\boldsymbol{Y}|\boldsymbol{X}+Q\boldsymbol{X}}|} \\
&= \frac{1}{2} \log \frac{|\boldsymbol{\Sigma}_{\boldsymbol{Y}} - \boldsymbol{\Sigma}_{\boldsymbol{YX}}\boldsymbol{\Sigma}_{\boldsymbol{X}}^{-1}\boldsymbol{\Sigma}_{\boldsymbol{XY}}|}{|\boldsymbol{\Sigma}_{\boldsymbol{Y}} - \boldsymbol{\Sigma}_{\boldsymbol{YX}}\boldsymbol{\Sigma}_{\boldsymbol{X}}^{-1}\boldsymbol{\Sigma}_{\boldsymbol{XY}}|} \\
&= 0
\end{aligned}
\tag{48}
$$

The entropy change of 0 indicates that the implementation of linear transformation to the original images could not reduce the complexity of the task.

### 1.3 INFLUENCE OF SALT-AND-PEPPER NOISE ON TASK ENTROPY

Salt-and-pepper noise is a common type of noise that can occur in images due to various factors, such as signal transmission errors, faulty sensors, or other environmental factors Chan et al. (2005). Salt-and-pepper noise is often considered to be an independent process because it is a type of random noise that affects individual pixels in an image independently of each other Gonzales & Wintz (1987).

### 1.3.1 INJECT SALT-AND-PEPPER NOISE IN LATENT SPACE

The entropy change of injecting salt-and-pepper noise can be formulated as:

$$
\begin{aligned}
\triangle S(\mathcal{T}, Q\boldsymbol{X}) =& H(\mathcal{T}; \boldsymbol{X}) - H_L(\mathcal{T}; \boldsymbol{X\epsilon}) \\
=& H(\boldsymbol{Y}; \boldsymbol{X}) - H(\boldsymbol{X}) - (H(\boldsymbol{Y}; \boldsymbol{X\epsilon}) - H(\boldsymbol{X})) \\
=& H(\boldsymbol{Y}; \boldsymbol{X}) - H(\boldsymbol{Y}; \boldsymbol{X\epsilon}) \\
=& -\sum_{\boldsymbol{X}\in\mathcal{X}}\sum_{\boldsymbol{Y}\in\mathcal{Y}} p(\boldsymbol{X},\boldsymbol{Y})\log p(\boldsymbol{X},\boldsymbol{Y}) + \sum_{\boldsymbol{X}\in\mathcal{X}}\sum_{\boldsymbol{Y}\in\mathcal{Y}}\sum_{\boldsymbol{\epsilon}\in\mathcal{E}} p(\boldsymbol{X\epsilon},\boldsymbol{Y})\log p(\boldsymbol{X\epsilon},\boldsymbol{Y}) \\
=& \mathbb{E}\left[\log\frac{1}{p(\boldsymbol{X},\boldsymbol{Y})}\right] - \mathbb{E}\left[\log\frac{1}{p(\boldsymbol{X\epsilon},\boldsymbol{Y})}\right] \\
=& \mathbb{E}\left[\log\frac{1}{p(\boldsymbol{X},\boldsymbol{Y})}\right] - \mathbb{E}\left[\log\frac{1}{p(\boldsymbol{X},\boldsymbol{Y})}\right] - \mathbb{E}\left[\log\frac{1}{p(\boldsymbol{\epsilon})}\right] \\
=& -\mathbb{E}\left[\log\frac{1}{p(\boldsymbol{\epsilon})}\right] \\
=& -H(\boldsymbol{\epsilon})
\end{aligned}
\tag{49}
$$

The entropy change is smaller than 0, therefore, the salt-and-pepper is a pure detrimental noise to the learning task.

### 1.3.2 ADD SALT-AND-PEPPER NOISE TO ORIGINAL IMAGES

The task entropy with salt-and-pepper noise is rewritten as:

$$
H(\mathcal{T}; \boldsymbol{X\epsilon}) = -\sum_{\boldsymbol{Y}\in\mathcal{Y}} p(\boldsymbol{Y}|\boldsymbol{X\epsilon})\log p(\boldsymbol{Y}|\boldsymbol{X\epsilon})
\tag{50}
$$

Since $\boldsymbol{\epsilon}$ is independent of $\boldsymbol{X}$ and $\boldsymbol{Y}$, the above equation can be expanded as:

$$
\begin{aligned}
H(\mathcal{T}; \boldsymbol{X\epsilon}) =& -\sum_{\boldsymbol{Y}\in\mathcal{Y}} \frac{p(\boldsymbol{Y},\boldsymbol{X\epsilon})}{p(\boldsymbol{X})p(\boldsymbol{\epsilon})}\log\frac{p(\boldsymbol{Y},\boldsymbol{X\epsilon})}{p(\boldsymbol{X})p(\boldsymbol{\epsilon})} \\
=& -\sum_{\boldsymbol{Y}\in\mathcal{Y}} \frac{p(\boldsymbol{Y},\boldsymbol{X})p(\boldsymbol{\epsilon})}{p(\boldsymbol{X})p(\boldsymbol{\epsilon})}\log\frac{p(\boldsymbol{Y},\boldsymbol{X})p(\boldsymbol{\epsilon})}{p(\boldsymbol{X})p(\boldsymbol{\epsilon})} \\
=& -\sum_{\boldsymbol{Y}\in\mathcal{Y}} p(\boldsymbol{Y}|\boldsymbol{X})\log p(\boldsymbol{Y}|\boldsymbol{X})
\end{aligned}
\tag{51}
$$

where

$$
\begin{aligned}
p(\boldsymbol{X\epsilon},\boldsymbol{Y}) =& p(\boldsymbol{X\epsilon}|\boldsymbol{Y})p(\boldsymbol{Y}) \\
=& p(\boldsymbol{X}|\boldsymbol{Y})p(\boldsymbol{\epsilon}|\boldsymbol{Y})p(\boldsymbol{Y}) \\
=& p(\boldsymbol{X}|\boldsymbol{Y})p(\boldsymbol{\epsilon})p(\boldsymbol{Y}) \\
=& p(\boldsymbol{X},\boldsymbol{Y})p(\boldsymbol{\epsilon})
\end{aligned}
\tag{52}
$$

Therefore, the entropy change with salt-and-pepper noise is:

$$
\triangle S(\mathcal{T}, Q\boldsymbol{X}) = H(\mathcal{T}; \boldsymbol{X}) - H(\mathcal{T}; \boldsymbol{X\epsilon}) = 0
\tag{53}
$$

Salt-and-pepper noise can not help reduce the complexity of the task, and therefore, it is considered a type of pure detrimental noise.

From the discussion in this section, we can draw conclusions that **Linear Transform Noise** can be positive under certain conditions, while **Gaussian Noise** and **Salt-and-pepper Noise** are harmful noise. From the above analysis, the conditions that satisfy positive noise are forming a design space. Exploring the positive noise space is an important topic for future work.

## 2 OPTIMAL QUALITY MATRIX OF LINEAR TRANSFORM NOISE

The optimal quality matrix should maximize the entropy change that leads toward the minimum of the task complexity. The optimization problem as formulated in Equation 37 is:

$$
\begin{aligned}
&\max_Q - \log |I + Q| \\
&s.t.\ rank(I + Q) = k \\
&\quad Q \sim I \\
&\quad [I + Q]_{ii} \geq [I + Q]_{ij}, i \neq j \\
&\quad \|[I + Q]_i\|_1 = 1
\end{aligned}
\tag{54}
$$

Maximizing the entropy change is to minimize the determinant of the matrix sum of $I$ and $Q$. A simple but straight way is to design the matrix $Q$ that makes the elements in $I + Q$ equal, i.e.,

$$
I + Q = \begin{bmatrix} 1/k & \cdots & 1/k \\ \vdots & \ldots & \vdots \\ 1/k & \cdots & 1/k \end{bmatrix}
\tag{55}
$$

The determinant of the above equation is 0, but it breaks the first constraint of $rank(I + Q) = k$. However, by adding a small constant into the diagonal, and minus another constant by other elements, we can get:

$$
I + Q = \begin{bmatrix} 1/k + c_1 & \cdots & & 1/k - c_2 \\ 1/k - c_2 & \ddots & & \vdots \\ \vdots & & \ddots & 1/k - c_2 \\ 1/k - c_2 & \cdots & 1/k - c_2 & 1/k + c_1 \end{bmatrix}
\tag{56}
$$

Under the constraints, we can obtain the two constants that fulfill the requirements:

$$
c_1 = \frac{k-1}{k(k+1)}, \quad c_2 = \frac{1}{k(k+1)}
\tag{57}
$$

Therefore, the corresponding $Q$ is:

$$
Q_{optimal} = \mathrm{diag}\left(\frac{1}{k+1} - 1, \ldots, \frac{1}{k+1} - 1\right) + \frac{1}{k+1}\mathbf{1}_{k \times k}
\tag{58}
$$

and the corresponding $I + Q$ is:

$$
I + Q = \begin{bmatrix} 2/(k+1) & \cdots & & 1/(k+1) \\ 1/(k+1) & \ddots & & \vdots \\ \vdots & & \ddots & 1/(k+1) \\ 1/(k+1) & \cdots & 1/(k+1) & 2/(k+1) \end{bmatrix}
\tag{59}
$$

As a result, the determinant of optimal $I + Q$ can be obtained by following the identical procedure as Equation 21:

$$
|I + Q| = \frac{1}{(k+1)^{k-1}}
\tag{60}
$$

The upper boundary of entropy change, i.e., the minimum of the task complexity of linear transform noise is determined:

$$
\triangle S(\mathcal{T}, Q\boldsymbol{X})_{upper} = (k-1)\log(k+1)
\tag{61}
$$

## 3 EXPERIMENTAL SETTING

We introduce the implementation details in this part. Model details are shown in Table 1 and 2. The image resolution is $224 \times 224$ for all the experiments. Pre-trained models on ImageNet are used as the backbone. We train all ResNet and ViT-based models using AdamW optimizer Loshchilov & Hutter (2017). We set the learning rate of each parameter group using a cosine annealing schedule

Table 1: Details of ResNet Models. The columns "18-layer", "34-layer", "50-layer", and "101-layer" show the specifications of ResNet-18, ResNet-34, ResNet-50, and ResNet-101, separately.

| Layer name | Output size | 18-layer | 34-layer | 50-layer | 101-layer |
|---|---|---|---|---|---|
| conv1 | $112 \times 112$ | $7 \times 7$, 64, stride 2 | | | |
| | | $3 \times 3$, max pool, stride 2 | | | |
| conv2_x | $56 \times 56$ | $\begin{bmatrix} 3 \times 3 & 64 \\ 3 \times 3 & 64 \end{bmatrix} \times 2$ | $\begin{bmatrix} 3 \times 3 & 64 \\ 3 \times 3 & 64 \end{bmatrix} \times 3$ | $\begin{bmatrix} 1 \times 1 & 64 \\ 3 \times 3 & 64 \\ 1 \times 1 & 256 \end{bmatrix} \times 3$ | $\begin{bmatrix} 1 \times 1 & 64 \\ 3 \times 3 & 64 \\ 1 \times 1 & 256 \end{bmatrix} \times 3$ |
| conv3_x | $28 \times 28$ | $\begin{bmatrix} 3 \times 3 & 128 \\ 3 \times 3 & 128 \end{bmatrix} \times 2$ | $\begin{bmatrix} 3 \times 3 & 128 \\ 3 \times 3 & 128 \end{bmatrix} \times 4$ | $\begin{bmatrix} 1 \times 1 & 128 \\ 3 \times 3 & 128 \\ 1 \times 1 & 512 \end{bmatrix} \times 4$ | $\begin{bmatrix} 1 \times 1 & 128 \\ 3 \times 3 & 128 \\ 1 \times 1 & 512 \end{bmatrix} \times 4$ |
| conv4_x | $14 \times 14$ | $\begin{bmatrix} 3 \times 3 & 256 \\ 3 \times 3 & 256 \end{bmatrix} \times 2$ | $\begin{bmatrix} 3 \times 3 & 256 \\ 3 \times 3 & 256 \end{bmatrix} \times 6$ | $\begin{bmatrix} 1 \times 1 & 256 \\ 3 \times 3 & 256 \\ 1 \times 1 & 1024 \end{bmatrix} \times 6$ | $\begin{bmatrix} 1 \times 1 & 256 \\ 3 \times 3 & 256 \\ 1 \times 1 & 1024 \end{bmatrix} \times 23$ |
| conv5_x | $7 \times 7$ | $\begin{bmatrix} 3 \times 3 & 512 \\ 3 \times 3 & 512 \end{bmatrix} \times 2$ | $\begin{bmatrix} 3 \times 3 & 512 \\ 3 \times 3 & 512 \end{bmatrix} \times 3$ | $\begin{bmatrix} 1 \times 1 & 512 \\ 3 \times 3 & 512 \\ 1 \times 1 & 2048 \end{bmatrix} \times 3$ | $\begin{bmatrix} 1 \times 1 & 512 \\ 3 \times 3 & 512 \\ 1 \times 1 & 2048 \end{bmatrix} \times 3$ |
| | $1 \times 1$ | average pool, 1000-d fc, softmax | | | |
| Params | | 11M | 22M | 26M | 45M |

Table 2: Details of ViT Models. Each row shows the specifications of a kind of ViT model. ViT-T, ViT-S, ViT-B, and ViT-L represent ViT Tiny, ViT Small, ViT Base, and ViT Large, separately.

| ViT Model | Layers | Hidden size | MLP size | Heads | Params |
|---|---|---|---|---|---|
| ViT-T | 12 | 192 | 768 | 3 | 5.7M |
| ViT-S | 12 | 384 | 1536 | 6 | 22M |
| ViT-B | 12 | 768 | 3072 | 12 | 86M |
| ViT-L | 12 | 1024 | 4096 | 16 | 307M |

Table 3: Variants of ViT with different kinds of noise on TinyImageNet. Vanilla means the vanilla model without noise. Accuracy is shown in percentage. Gaussian noise used here is subjected to standard normal distribution. Linear transform noise used in this table is designed to be positive noise. The difference is shown in the bracket.

| Model | DeiT | SwinTransformer | BeiT | ConViT |
|---|---|---|---|---|
| Vanilla | 85.02 (+0.00) | 90.84 (+0.00) | 88.64 (+0.00) | 90.69 (+0.00) |
| + Gaussian Noise | 84.70 (-0.32) | 90.34 (-0.50) | 88.40 (-0.24) | 90.40 (-0.29) |
| + Linear Transform Noise | **86.50 (+1.48)** | **95.68 (+4.84)** | **91.78 (+3.14)** | **93.07 (+2.38)** |
| + Salt-and-pepper Noise | 84.03 (-1.01) | 87.12 (-3.72) | 42.18 (-46.46) | 89.93 (-0.76) |
| Params. | 86M | 87M | 86M | 86M |

Table 4: ResNet with different kinds of noise on TinyImageNet. Vanilla means the vanilla model without noise. Accuracy is shown in percentage. Gaussian noise used here is subjected to standard normal distribution. Linear transform noise used in this table is designed to be positive noise. The difference is shown in the bracket.

| Model | ResNet-18 | ResNet-34 | ResNet-50 | ResNet-101 |
|---|---|---|---|---|
| Vanilla | 64.01 (+0.00) | 67.04 (+0.00) | 69.47 (+0.00) | 70.66 (+0.00) |
| + Gaussian Noise | 63.23 (-0.78) | 65.71 (-1.33) | 68.17 (-1.30) | 69.13 (-1.53) |
| + Linear Transform Noise | **73.32 (+9.31)** | **76.70 (+9.66)** | **76.88 (+7.41)** | **77.30 (+6.64)** |
| + Salt-and-pepper Noise | 55.97 (-8.04) | 63.52 (-3.52) | 49.42 (-20.25) | 53.88 (-16.78) |

Table 5: ViT with different kinds of noise on TinyImageNet. Vanilla means the vanilla model without injecting noise. Accuracy is shown in percentage. Gaussian noise used here is subjected to standard normal distribution. Linear transform noise used in this table is designed to be positive noise. The difference is shown in the bracket. Note **ViT-L is overfitting on TinyImageNet** Dosovitskiy et al. (2020) Steiner et al. (2021).

| Model | ViT-T | ViT-S | ViT-B | ViT-L |
|---|---|---|---|---|
| Vanilla | 81.75 (+0.00) | 86.78 (+0.00) | 90.48 (+0.00) | 93.32 (+0.00) |
| + Gaussian Noise | 80.95 (-0.80) | 85.66 (-1.12) | 89.61 (-0.87) | 92.31 (-1.01) |
| + Linear Transform Noise | **82.50 (+0.75)** | **91.62 (+4.84)** | **94.92 (+4.44)** | **93.63 (+0.31)** |
| + Salt-and-pepper Noise | 79.34 (-2.41) | 84.66 (-2.12) | 87.45 (-3.03) | 83.48 (-9.84) |

Table 6: Top 1 accuracy on ImageNet V2 with positive linear transform noise.

| Model | Top1 Acc. | Params. | Image Res. | Pretrained Dataset |
|---|---|---|---|---|
| ViT-B/16-SAM | 67.50 | 86M | $224 \times 224$ | ImageNet 21k |
| LeViT-256 | 69.90 | 86M | $224 \times 224$ | ImageNet 21k |
| LeViT-384 | 71.40 | 86M | $384 \times 384$ | ImageNet 21k |
| ViT-B+PN | **82.23** | 86M | $224 \times 224$ | ImageNet 21k |
| ViT-B+PN | **84.85** | 86M | $384 \times 384$ | ImageNet 21k |

with a minimum of $1e - 7$. The data augmentation for training only includes the random resized crop and normalization.

**CNN(ResNet) Setting** The training epoch is set to 100. We initialized the learning rate as 0 and linearly increase it to 0.001 after 10 warmup steps. All the experiments of CNNs are trained on a single Tesla V100 GPU with 32 GB. The batch size for ResNet18, ResNet34, ResNet50, and ResNet101 are 1024, 512, 256, and 128, respectively.

**ViT and Variants Setting** All the experiments of ViT and its variants are trained on a single machine with 8 Tesla V100 GPUs. For vanilla ViTs, including ViT-T, ViT-S, ViT-B, and ViT-L, the training epoch is set to 50 and the input patch size is $16 \times 16$. We initialized the learning rate as 0 and linearly increase it to 0.0001 after 10 warmup steps. We then decrease it by the cosine decay strategy. For experiments on the variants of ViT, the training epoch is set to 100 and the learning rate is set to 0.0005 with 10 warmup steps.

## 4 MORE EXPERIMENT RESULTS

We show more experiment results of injecting positive noise to other variants of the ViT family, such as SwinTransformer, DeiT, ConViT, and BeiT, and implement them on the smaller dataset, i.e., TinyImageNet. Note, considering limited computational resources, all the experiments in the supplementary are on the TinyImageNet. The strength of positive noise is set to 0.3. The noise is injected into the last layer.

### 4.1 INJECT POSITIVE NOISE TO VARIANTS OF VIT

As demonstrated in the paper, the positive noise can be injected into the ViT family. Therefore, in this section, we explore the influence of positive noise on the variants of the ViT. The positive noise used here is identical to that in the paper. For this, we comprehensively compare noise injection to ConViT d'Ascoli et al. (2021), BeiT Bao et al. (2021), DeiT Touvron et al. (2021), and Swin Transformer Liu et al. (2021), and comparisons results are reported in Tabel 3. As expected, these variants of ViTs get benefit from the positive noise. The additional four ViT variants are at the base scale, whose parameters are listed in the table's last row. For a fair comparison, we use identical experimental settings for each kind of experiment. For example, we use the identical setting for vanilla ConViT, ConViT with different kinds of noise. From the experimental results, we can observe that the different variants of ViT benefit from positive noise and significantly improve prediction accuracy. The results on different scale datasets and variants of the ViT family demonstrate that positive noise can universally improve the model performance by a wide margin.

Table 7: Comparison with SOTA methods on **Office-Home**. The best performance is marked in red.

| Method | Ar→Cl | Ar→Pr | Ar→Re | Cl→Ar | Cl→Pr | Cl→Re | Pr→Ar | Pr→Cl | Pr→Re | Re→Ar | Re→Cl | Re→Pr | Avg. |
|---|---|---|---|---|---|---|---|---|---|---|---|---|---|
| ResNet-50 | 44.9 | 66.3 | 74.3 | 51.8 | 61.9 | 63.6 | 52.4 | 39.1 | 71.2 | 63.8 | 45.9 | 77.2 | 59.4 |
| MinEnt | 51.0 | 71.9 | 77.1 | 61.2 | 69.1 | 70.1 | 59.3 | 48.7 | 77.0 | 70.4 | 53.0 | 81.0 | 65.8 |
| SAFN | 52.0 | 71.7 | 76.3 | 64.2 | 69.9 | 71.9 | 63.7 | 51.4 | 77.1 | 70.9 | 57.1 | 81.5 | 67.3 |
| CDAN+E | 54.6 | 74.1 | 78.1 | 63.0 | 72.2 | 74.1 | 61.6 | 52.3 | 79.1 | 72.3 | 57.3 | 82.8 | 68.5 |
| DCAN | 54.5 | 75.7 | 81.2 | 67.4 | 74.0 | 76.3 | 67.4 | 52.7 | 80.6 | 74.1 | 59.1 | 83.5 | 70.5 |
| BNM | 56.7 | 77.5 | 81.0 | 67.3 | 76.3 | 77.1 | 65.3 | 55.1 | 82.0 | 73.6 | 57.0 | 84.3 | 71.1 |
| SHOT | 57.1 | 78.1 | 81.5 | 68.0 | 78.2 | 78.1 | 67.4 | 54.9 | 82.2 | 73.3 | 58.8 | 84.3 | 71.8 |
| ATDOC-NA | 58.3 | 78.8 | 82.3 | 69.4 | 78.2 | 78.2 | 67.1 | 56.0 | 82.7 | 72.0 | 58.2 | 85.5 | 72.2 |
| ViT-B | 54.7 | 83.0 | 87.2 | 77.3 | 83.4 | 85.6 | 74.4 | 50.9 | 87.2 | 79.6 | 54.8 | 88.8 | 75.5 |
| TVT-B | 74.9 | 86.8 | 89.5 | 82.8 | 88.0 | 88.3 | 79.8 | 71.9 | 90.1 | 85.5 | 74.6 | 90.6 | 83.6 |
| CDTrans-B | 68.8 | 85.0 | 86.9 | 81.5 | 87.1 | 87.3 | 79.6 | 63.3 | 88.2 | 82.0 | 66.0 | 90.6 | 80.5 |
| SSRT-B | 75.2 | 89.0 | 91.1 | 85.1 | 88.3 | 90.0 | 85.0 | 74.2 | 91.3 | 85.7 | 78.6 | 91.8 | 85.4 |
| TVT-B+PN | **78.3** | **90.6** | **91.9** | **87.8** | **92.1** | **91.9** | **85.8** | **78.7** | **93.0** | **88.6** | **80.6** | **93.5** | **87.7** |

Table 8: Comparison with SOTA methods on **Visda2017**. The best performance is marked in red.

| Method | plane | bcycl | bus | car | horse | knife | mcycl | person | plant | sktbrd | train | truck | Avg. |
|---|---|---|---|---|---|---|---|---|---|---|---|---|---|
| ResNet-50 | 55.1 | 53.3 | 61.9 | 59.1 | 80.6 | 17.9 | 79.7 | 31.2 | 81.0 | 26.5 | 73.5 | 8.5 | 52.4 |
| DANN | 81.9 | 77.7 | 82.8 | 44.3 | 81.2 | 29.5 | 65.1 | 28.6 | 51.9 | 54.6 | 82.8 | 7.8 | 57.4 |
| MinEnt | 80.3 | 75.5 | 75.8 | 48.3 | 77.9 | 27.3 | 69.7 | 40.2 | 46.5 | 46.6 | 79.3 | 16.0 | 57.0 |
| SAFN | 93.6 | 61.3 | 84.1 | 70.6 | 94.1 | 79.0 | 91.8 | 79.6 | 89.9 | 55.6 | 89.0 | 24.4 | 76.1 |
| CDAN+E | 85.2 | 66.9 | 83.0 | 50.8 | 84.2 | 74.9 | 88.1 | 74.5 | 83.4 | 76.0 | 81.9 | 38.0 | 73.9 |
| BNM | 89.6 | 61.5 | 76.9 | 55.0 | 89.3 | 69.1 | 81.3 | 65.5 | 90.0 | 47.3 | 89.1 | 30.1 | 70.4 |
| CGDM | 93.7 | 82.7 | 73.2 | 68.4 | 92.9 | 94.5 | 88.7 | 82.1 | 93.4 | 82.5 | 86.8 | 49.2 | 82.3 |
| SHOT | 94.3 | 88.5 | 80.1 | 57.3 | 93.1 | 93.1 | 80.7 | 80.3 | 91.5 | 89.1 | 86.3 | 58.2 | 82.9 |
| ViT-B | 97.7 | 48.1 | 86.6 | 61.6 | 78.1 | 63.4 | 94.7 | 10.3 | 87.7 | 47.7 | 94.4 | 35.5 | 67.1 |
| TVT-B | 92.9 | 85.6 | 77.5 | 60.5 | 93.6 | 98.2 | 89.4 | 76.4 | 93.6 | 92.0 | 91.7 | 55.7 | 83.9 |
| CDTrans-B | 97.1 | 90.5 | 82.4 | 77.5 | 96.6 | 96.1 | 93.6 | **88.6** | **97.9** | 86.9 | 90.3 | 62.8 | 88.4 |
| SSRT-B | **98.9** | 87.6 | **89.1** | **84.8** | 98.3 | **98.7** | **96.3** | 81.1 | 94.9 | 97.9 | 94.5 | 43.1 | 88.8 |
| TVT-B+PN | 98.8 | **95.5** | 84.8 | 73.7 | **98.5** | 97.2 | 95.1 | 76.5 | 95.9 | **98.4** | **98.3** | **67.2** | **90.0** |

## 4.2 Positive Noise on TinyImageNet

We also implement experiments of ResNet and ViT on the smaller dataset TinyImageNet, and the results are shown in Table 4 and 5. As shown in the tables, positive noise also benefits the deep models on the small dataset. From the experiment results of CNN and ViT family on ImageNet and TinyImageNet, we can find that the positive noise has better effects on larger datasets than smaller ones. This makes sense because as shown in the section on optimal quality matrix, the upper boundary of the entropy change is determined by the size, i.e., the number of data samples, of the dataset, smaller datasets have less number of data samples, which means the upper boundary of the small datasets is lower than the large datasets. Therefore, the positive noise of linear transform noise has better influences on large than small datasets.

## 4.3 Experiments in ImageNet V2

We also implement additional experiments on ImageNet V2. The results are shown in Table 6. Compared to existing methods based on ViT with similar parameters, the ViT enhanced by positive noise demonstrates a significant improvement in performance.

## 4.4 Positive Noise for Domain Adaptation

Unsupervised domain adaptation (UDA) aims to learn transferable knowledge across the source and target domains with different distributions Pan & Yang (2009) Wei et al. (2018). There are mainly two kinds of deep neural networks for UDA, which are CNN-based and Transformer-based methods Sun et al. (2022) Yang et al. (2023). Various techniques for UDA are adopted on these

backbone architectures. For example, the discrepancy techniques measure the distribution divergence between source and target domains Long et al. (2018) Sun & Saenko (2016). Adversarial adaptation discriminates domain-invariant and domain-specific representations by playing an adversarial game between the feature extractor and a domain discriminator Ganin & Lempitsky (2015).

Recently, transformer-based methods achieved SOTA results on UDA, therefore, we evaluate the ViT-B with the positive noise on widely used UDA benchmarks. Here the positive noise is the linear transform noise identical to that used in the classification task. The positive noise is injected into the last layer of the model, the same as the classification task. The datasets include **Office Home** Venkateswara et al. (2017) and **VisDA2017** Peng et al. (2017). **Office-Home**Venkateswara et al. (2017) has 15,500 images of 65 classes from four domains: Artistic (Ar), Clip Art (Cl), Product (Pr), and Real-world (Rw) images. **VisDA2017** is a Synthetic-to-Real object recognition dataset, with more than 0.2 million images in 12 classes. We use the ViT-B with a $16 \times 16$ patch size, pre-trained on ImageNet. We use minibatch Stochastic Gradient Descent (SGD) optimizer Ruder (2016) with a momentum of 0.9 as the optimizer. The batch size is set to 32. We initialized the learning rate as 0 and linearly warm up to 0.05 after 500 training steps. The results are shown in Table 7 and 8. The methods above the black line are based on CNN architecture, while those under the black line are developed from the Transformer architecture. The ViT-B with positive noise achieves better performance than the existing works. These results show that positive noise can improve model generality, therefore, benefit deep models in domain adaptation tasks.