# OpenReview forum: "Exploring the Impact of Information Entropy Change in Learning Systems"
_ICLR.cc/2024/Conference — Submitted to ICLR 2024_

### Official Review · Reviewer_FieS · 2023-10-31

**Soundness:** 2 fair
**Presentation:** 3 good
**Contribution:** 1 poor
**Rating:** 3
**Confidence:** 5

**Summary:**

The paper aims to explore the influence of the information entropy change, and specifically analyzes the impact of various types of noise, i.e., Gaussian noise, linear transform noise, and salt-and-pepper noise, on the performance of deep learning models for image classification and domain adaptation tasks. The authors verify their method on different network architectures, i.e., CNNs and ViTs, and show that the positive noise injection can improve the accuracy of image classification.

**Strengths:**

1. The proposed method of using noise injection to improve the performance of CNNs and ViTs, which has not been extensively explored before.

2. The article is well-written and easy to understand, with clear explanations of the proposed method and the experimental results.

**Weaknesses:**

1.Limited analysis between data augmentation and the proposed noise injection. It seems more like a feature augmentation method for different layers. The article claims that the positive noise is benefit of the classification network, but it is hard to measure the noise level for the positive/negative influence. It is encouraged to compare the difference between the data augmentation with noise and the proposed method.

2.Noise definition confusion. Unlike Gaussian noise and salt-and-pepper noise with the specific probability distribution, the linear transform is not a kind of noise, but a simple operation.

3.Limited analysis of the tasks. The authors leverage the proposed method in image classification tasks, including domain adaptation. It is encouraged to conduct the noise injection in image object detection/segmentation tasks, or some NLP tasks, to verify the effectiveness of the proposed method.

**Questions:**

See weaknesses.

---

> ### Author Response · Authors · 2023-11-14
>
> Question 1. Limited analysis between data augmentation and the proposed noise injection. It seems more like a feature augmentation method for different layers. The article claims that the positive noise is benefit of the classification network, but it is hard to measure the noise level for the positive/negative influence. It is encouraged to compare the difference between the data augmentation with noise and the proposed method.
>
> Our answer:
> Thanks for your questions. The information entropy of the learning system framework proposed in our works provides theoretical explanations for data augmentation.
>
> Data augmentation can be divided into two aspects:
>
> Image Level Data Augmentation, such as flips, translations, rotations, CutOut, MixUp, CutMix, and etc.
> As the reviewer eupP already pointed out, the pioneering work on Positive-Incentive Noise (IEEE Transactions on Neural Networks and Learning Systems, 2022) has provided a theoretical framework for image-level data augmentation.
>
> Latent Data Augmentation, such as adding noise to embeddings, feature mixup, and etc.
> Our work provided a theoretical framework for latent data augmentation.
>
> Therefore, our work provides theoretical guidance for exploring possible effective data augmentation methods in the future.
> Our work provides quantitative tools for analyzing the influence of adding noise to learning systems. Combined with the previous work on Positive-Incentive Noise in IEEE TNNLS 2022, readers can analyze the effects of adding noise at both the image and latent levels.
>
> Question 2. Noise definition confusion. Unlike Gaussian noise and salt-and-pepper noise with the specific probability distribution, the linear transform is not a kind of noise, but a simple operation.
>
> Our answer: Thank you for bringing this up. For a specific image X1, Gaussian noise, salt-and-pepper noise are considered noise; this is no doubt for reviewer FieS. However, in our definition, we consider another image X2 as noise when it is added to X1. We will emphasize this point if there is confusion for the reviewer.
>
> Question 3. Limited analysis of the tasks. The authors leverage the proposed method in image classification tasks, including domain adaptation. It is encouraged to conduct the noise injection in image object detection/segmentation tasks, or some NLP tasks, to verify the effectiveness of the proposed method.
>
> Our answer: Great question. Your suggestions are the directions we will work on, but our expertise lies in image classification, domain adaptation, and semi-supervised learning. We may need collaborators to expand our research. For NLP tasks, there was an arXiv submission last month that utilized uniform noise as positive noise, where they significantly improved instruction fine-tuning in large language models. For object detection/segmentation, we are interested in applying positive noise in these areas, but we need time to delve into it.

---

### Official Review · Reviewer_eupP · 2023-11-01

**Soundness:** 3 good
**Presentation:** 3 good
**Contribution:** 3 good
**Rating:** 8
**Confidence:** 3

**Summary:**

This paper examines the impact of task entropy change in deep neural networks by introducing noise at different levels in a network, and demonstrates that certain kinds of noise can actually help with learning by reducing the task entropy. The paper differentiates between "positive noise" (PN) that can enhance system performance by reducing task complexity and "harmful noise" (HN) that deteriorates it. The concept of "positive noise" is introduced in [1] which this work cites. By using information entropy as a measure of task complexity, the research shows that intentionally adding positive noise can substantially improve performance. The empirical findings challenge traditional views on noise in deep learning, suggesting that positive noise can be beneficial. Results are demonstrated using networks from the ResNet and ViT family for classification on ImageNet and using ViT-B for unsupervised domain adaptation on Office-Home and Visda2017 datasets.

[1] Xuelong Li. Positive-incentive noise. IEEE Transactions on Neural Networks and Learning Systems,
2022.

**Strengths:**

The paper is well written. The supplementary section provides further details on the derivations, which is very helpful.

The technique is well motivated and the empirical results are quite impressive.

This work provides an interesting new perspective on positive noise injection which can help training.

**Weaknesses:**

No error bars in any tables - did the authors run multiple seeds for their experiments? Even though the improvements are generally large, it is nice to see these in the tables/figures.

The technique is only evaluated on classification using ImageNet and domain adaptation. It would be good to see results on other tasks like object detection perhaps, and specially domains like Natural Language Processing, which differ from vision based tasks.

**Questions:**

Given that the improvement is so large on ResNet-18, do the authors have an explanation for why ViT-T does not improve similarly?

Can the authors elaborate on this statement? "Besides, when the models are corrupted under brute force attack, the positive noise also can not work." Why is this the case?

This point also requires more detailed explanation - "Second, injection to shallow layers obtain less entropy change gain because of trendy replacing Equation 8 with Equation 7." The explanation for why shallower layers don't provide as high accuracy gains as deeper layers can be improved.

Have the authors tried combinations of later layers for noise injection?

Given that this is a new perspective and the results are strong, will code be released for reproducibility?

Can the authors give other examples of positive noise that they considered?

---

> ### Author Response · Authors · 2023-11-14
>
> Thank you for your constructive feedback and valuable comments. We are happy to answer your questions.
>
> Answer for Weakness1:
> Thank you for your inquiry regarding the technique details. To ensure a fair comparison in experiments with different models on ImageNet, we maintain consistency by using seed_everything with an identical seed in each experiment. Notably, given the substantial improvements that significantly surpass the SoTA, we let two colleagues independently conducted the experiments on different machines, and the reported results for ImageNet are averaged. However, due to limited computational resources, other experiments were run only once. We appreciate your suggestion, and in response, we plan to compile all experiment information, presenting it in a figure within the supplementary material.
>
> Answer for Weakness2:
> Great suggestions. We have found other works submitted on ArXiv in the last month that utilize positive noise to enhance instruction finetuning in large language models. At the very least, the application of positive noise to NLP has been verified.
> Since our expertise lies primarily in image classification, domain adaptation, and semi-supervised learning, we are actively seeking potential collaborators to apply our methodology in other areas across CV and NLP.  If the reviewer is an expert in related areas, we would be more than happy to discuss potential opportunities.
>
>
> Answer for Question1:
> The details of the model architectures, presented in Table 1 and 2 of the supplementary materials, may provide two key explanations.
>
> 1, the ViT-Tiny model has a head number of 3, significantly smaller than the 12 in
> ViT-B. This limitation hampers ViT-Tiny's ability to capture diverse patterns in images. This can be supported by the paper ‘On the Relationship between Self-Attention and Convolutional Layers’ ICLR 2020.
>
> 2, with only 5.7 million parameters, ViT-Tiny falls short of the parameter count in  ResNet-18 by nearly half. This suggests that ViT-Tiny may struggle to capture the necessary complexity and hierarchical features as effectively as ResNet-18 in ImageNet classification tasks.
>
>
> Answer for Question2:  The paragraph on the moderate model assumption claims that positive noise functions effectively for a 'normal' or 'ordinary' model. However, for instance, if a model is compromised, as demonstrated in the 'Color Backdoor: A Robust Poisoning Attack in Color Space,' CVPR 2023, the positive noise may not work. If this statement makes readers confused, we will revise it.
>
>
> Answer for Question3: Thanks for the great suggestions. An additional explanation could be that deeper layers lead to better representations. Evidence from 'Do Wide and Deep Networks Learn the Same Things? Uncovering How Neural Network Representations Vary with Width and Depth' (ICLR 2021) suggests that as the depth or width of the model increases, there is an emergence of principal components that contribute to improved performance. This implies that a certain depth is necessary for the model to achieve satisfactory performance.
>
>
> Answer for Question4: No, all the experiments involve a single layer of noise injection. If we understand correctly, did the reviewer suggest trying noise injection into multiple layers? If so, the first author can prepare additional experiments if necessary.
>
>
> Answer for Question5: All members have already agreed that all the source code running CNNs and ViTs on ImageNet, ImageNet V2, Office-Home, and Visda2017 will be released on GitHub upon acceptance. We will also provide the pretrained models.
>
>
> Answer for Question6: This is a great question! Yes. We will explore more kinds of positive noises. Another team has recently achieved promising results by using uniform noise as positive noise in an NLP task; their related submission is already on ArXiv.
>
> The work by Xuelong Li, 'Positive-Incentive Noise,' published in IEEE Transactions on Neural Networks and Learning Systems in 2022, is a pioneering effort that introduces positive noise in machine learning. We also recommend another paper that explores positive noise in signal processing: 'Exploring Positive Noise in Estimation Theory' by Radnosrati, Kamiar, Gustaf Hendeby, and Fredrik Gustafsson, published in IEEE Transactions on Signal Processing in 2020. We strongly believe that there is a great potential for positive noise in AI community.

---

> > ### Comment · Area_Chair_QsRT · 2023-11-20
> > **Signal-boosting**
> >
> > This is a friendly reminder to please read and respond to the authors' rebuttal in time for the end of the discussion period, which is this Wednesday, November 22. Thank you!
> >
> > Your Area Chair

---

> > ### Comment · Reviewer_eupP · 2023-11-20
> > **Thanks for your clarifications.**
> >
> > Thank you for the clarifications to my questions. Can you add a pointer to the use of uniform noise as positive noise in an NLP task that you talk about in your response? I would like to read it.
> >
> > For question 4, yes, I was asking about adding noise to more than one layers at a time as this would be an interesting experiment.

---

> > > ### Author Response · Authors · 2023-11-21
> > >
> > > 1. Regarding the usage of Positive Noise in NLP tasks, please refer to the paper at https://arxiv.org/pdf/2310.05914.pdf.
> > > 2. Thank you for the confirmation. We plan to conduct additional experiments of adding positive noise to different numbers of layers in ViTs and ResNets—ranging from two layers to three layers or more. The results of these experiments will be appended to the supplementary material. We appreciate your constructive feedback, and irrespective of the outcome of this work—whether accepted or rejected—we are committed to completing the experiments involving the addition of noise to multiple layers.

---

### Official Review · Reviewer_H7oc · 2023-11-01

**Soundness:** 3 good
**Presentation:** 3 good
**Contribution:** 3 good
**Rating:** 5
**Confidence:** 3

**Summary:**

This research investigates the impact of noise-induced entropy changes in deep learning systems, focusing on computer vision tasks. While noise is traditionally seen as detrimental, this study demonstrates that specific noise, termed positive noise (PN), can enhance deep learning model performance by reducing task complexity defined by information entropy. The study introduces a distinction between positive noise (beneficial) and harmful noise (detrimental) and shows that proactive injection of positive noise significantly improves accuracy, achieving over 95% on ImageNet. Besides, this paper explores three types of noises. But the difference of positive noises on different kinds of noise types lacks discussion.

**Strengths:**

Strengths:
- This paper challenges the notion that noise always hampers deep learning models, showcasing its potential as a positive influence.
- It offers theoretical insights, distinguishing noise impact at different levels, aiding in optimizing task complexity.
- Experiments on both CNNs and Vision Transformers are conducted.

**Weaknesses:**

- The range of tasks tackled remains relatively limited, lacking diversity and complexity in comparison to broader applications. To be specific, this paper only conducts experiments on classification tasks. More results on other tasks (e.g., regression tasks like Object Detection or generative tasks like language understanding)  are lacking to validate its generalization ability.

**Questions:**

For each type of noise, there will exist positive noises. What are their difference and influences?

---

> ### Author Response · Authors · 2023-11-14
>
> We appreciate your feedback and are dedicated to providing detailed answers to any questions.
>
> Weakness: The range of tasks tackled remains relatively limited, lacking diversity and complexity in comparison to broader applications. (e.g., regression tasks like Object Detection or generative tasks like language understanding)
>
> Our answer: We will certainly explore the tasks mentioned by the reviewer. For instance, the object detection task involves predicting both the coordinates of the bounding box and the label of the object. This differs from classification tasks, which only predict labels. Consequently, our plan includes conducting theoretical research on reducing information entropy in object detection tasks, followed by experimental validation.If the reviewer is an expert in object detection or related areas, there could be an opportunity for collaboration after the anonymous review period.
>
> Questions: For each type of noise, there will exist positive noises. What are their difference and influences?
>
> Our answer:
> According to the definition in formula (5), any noise that aids in reducing the entropy of the learning system is labeled as positive noise. The extent of performance improvement resulting from the addition of positive noise depends on the degree to which it decreases entropy. The greater the reduction in the system's entropy, the higher the expected increase in performance.
>
> In our view, these types of noises represent potential ways to alter the entropy of the learning system and influence its performance. One significant distinction is that salt-and-pepper and Gaussian noises have their own distributions and are independent of the data samples, whereas linear transform noise utilizes features from other data samples, making it dependent on the data samples.

---

> > ### Comment · Reviewer_H7oc · 2023-12-03
> >
> > Thanks for your reply. On the one hand, I think from the experimental perspective, the proposed method can be directly to object detection models, and the result is necessary to validate the method's effect. On the other hand, the results on the classification task are not reasonable since the ResNet-18 on ImageNet can reach 70\%. However, the vanilla results in Table 1 are obviously lower. The authors mentioned that the results are without data augmentation but this practice is not reasonable and should not bring such a significant performance degradation.

---

### Author Response · Authors · 2023-11-14
**General response**

We've observed that some reviewers express interest in applying our methodology to various tasks, including image segmentation, object detection, language understanding, and other learning applications within computer vision (CV) and natural language processing (NLP). These suggestions align well with our future research directions. However, we would like to emphasize that our primary expertise lies in image classification, domain adaptation, and semi-supervised learning within the field of CV. While we appreciate these valuable suggestions, we want to clarify that this work focuses specifically on image classification and domain adaptation. We remain open to addressing any questions within our expertise and are grateful for your constructive feedback.

We are honored that all reviewers recognize the applicability of the philosophy underpinning our paper across a broad spectrum of learning tasks in CV and NLP.

Re-emphasizing our contribution:
1. The alteration of information entropy through the addition of positive/harmful noise can be applied to mainstream model architectures, including CNNs, ViTs, and their variants such as ResNet, Wide ResNet, SwinTransformer, DeiT, ConViT, etc.
2. We have attained unparalleled performance, establishing new SoTA on both ImageNet and ImageNet V2, which are among the most renowned datasets in CV. Please consult the ranking list at https://paperswithcode.com/sota/image-classification-on-imagenet.
3. The success of positive noise in domain adaptation tasks demonstrates it also works in Out of Distribution (OOD) situations.
4. Our method offers a novel approach to enhancing model performance without requiring changes to model architectures or the creation of new loss functions.
5. The theory introduced in 'Positive-Incentive Noise' (IEEE TNNLS 2022) and the theory proposed in our work provides theoretical explanations for data augmentation, whether at the image level or in the latent space. Researchers can leverage the theory presented in this work to design effective data augmentation methods.

---

### Meta-Review · Area_Chair_QsRT · 2023-12-10

**Metareview:**

The submission looks into noise injection into deep neural networks from the perspective of its impact on entropy. The authors distinguish between "positive noise" (as defined by Li (2022)) and "harmful noise" and show that positive noise injection can improve model performance by reducing task complexity (defined relative to information entropy).

Reviewers note that the submission presents an interesting new perspective (eupP, FieS) that challenges a commonly held belief (H7oc, namely that noise always hampers deep learning models). They also note the paper's writing quality and clarity (eupP, FieS), its theoretical contributions (H7oc), and its convincing empirical results (eupP). All reviewers noted that the range of tasks considered is too narrow and should include regression and generative tasks. The authors leave this out for future work and clarify that their submission focuses specifically on image classification, domain adaptation, and semi-supervised learning within the field of computer vision.

Given the high variance in final scores across reviewers, I gave the submission a careful read. Below are my observations which complement the three review discussion threads in justifying my final recommendation.

- The most significant reviewer concern is the narrowness of the evaluation, and I think this is a fair objection: generalizability is an important property to assess, as there are so many confounders in machine learning problems that it's impossible to completely rule out that the proposed approach works (very well, I should note) for other reasons (which are tied to the nature of the computer vision problems being tackled) than the one claimed by the authors. The cited NLP paper goes some way towards alleviating this concern. It's understandable that the authors have their area of expertise and therefore are not in a position to evaluate their approach in every conceivable problem setting, especially within the timeframe of the rebuttal, however not all weaknesses identified by reviewers necessarily have to be fixable in the rebuttal period in order to be fair.
- I recognize this is a frustrating situation to be in, but I unfortunately can't take into account the authors' claim that other works incorporate their approach, as it can't be verified without breaking anonymity.
- The submission equates task complexity (as measured through information entropy) and task difficulty (as arguably measured via the ability of a model trained on that task to generalize) almost axiomatically ("For example, a smaller H(T ) means an easier task and vice versa."). Why would that be the case?
- It's unclear how to reconcile the theory's assumption that both X and Y can be made to follow an approximately normal distribution and the categorical distribution of Y in the image classification experiments. Also, the effect of latent noise injection on task complexity is never empirically verified; only its effect on classification performance is. How can we be sure that linear transform really did lower task complexity (as defined in the submission) in the experiments? I'm therefore not entirely convinced that the theory presented fully applies to the experiments presented.
- I don't completely agree with reviewers regarding the submission's clarity: (1) On implementation details, the supplementary material does offer lots of details, but at the same time the submission is unclear on when in the process the positive noise is injected. Is it during training? Is it also injected at inference time? It's not possible to tell from Figure 1, Section 5, or Appendix Section 3. (2) The mathematical notation looks overloaded: if epsilon is the additive or multiplicative noise injected in the latent space, how come epsilon is added to or multiplied with X in Equation 8 defining task complexity for latent noise injection?

**Justification For Why Not Higher Score:**

Overall, while the submission does present strong results, it does so in a narrowly-scoped application setting and does not convincingly establish a causal relationship between the theory presented and the strong results presented.

**Justification For Why Not Lower Score:**

N/A

---

### Decision · Program_Chairs · 2024-01-16

Reject